# Ecological and Syntaxonomic Analysis of the Communities of *Glebionis coronaria* and *G. discolor* (*Malvion neglectae*) in the European Mediterranean Area

**DOI:** 10.3390/plants13050568

**Published:** 2024-02-20

**Authors:** Eusebio Cano, Ana Cano-Ortiz, Ricardo Quinto Canas, Jose Carlos Piñar Fuentes, Catarina Rodrigues Meireles, Mauro Raposo, Carlos Pinto Gomes, Valentina Lucia Astrid Laface, Giovanni Spampinato, Carmelo Maria Musarella

**Affiliations:** 1Department of Animal and Plant Biology and Ecology, Section of Botany, University of Jaén, 23071 Jaén, Spain; jpinar@ujaen.es; 2Department of Didactics of Experimental, Social and Mathematical Sciences, Complutense University of Madrid, 28040 Madrid, Spain; acano07@ucm.es; 3Faculty of Sciences and Technology, University of Algarve, Campus de Gambelas, 8005-139 Faro, Portugal; rqcanas@gmail.com; 4Department of Landscape, Environment and Planning, Instituto Mediterrâneo para a Agricultura, Ambiente e Desenvolvimento (MED), Escola de Ciências e Tecnologia, University of Évora, 7004-516 Évora, Portugal; cmeireles@uevora.pt (C.R.M.); mraposo@uevora.pt (M.R.); cpgomes@uevora.pt (C.P.G.); 5Department of AGRARIA, “Mediterranea” University of Reggio Calabria, 89124 Reggio Calabria, Italy; vla.laface@unirc.it (V.L.A.L.); gspampinato@unirc.it (G.S.); carmelo.musarella@unirc.it (C.M.M.)

**Keywords:** new associations, nitrification, nomenclature, grasslands, sintaxonomy, taxonomy

## Abstract

Nitrophilous communities dominated by *Glebionis coronaria* and *Glebionis discolor* in the European Mediterranean area were studied. The nomenclature was corrected according to the current taxonomy, following the International Code of Phytosociological Nomenclature (ICPN). The statistical analysis revealed six new associations and one subassociation, with four in Spain, one in Greece, and one in Italy. Additionally, a subassociation of high relevance due to its endemic character was identified. These grasslands exhibit requirements for organic matter and other edaphic nutrients that are closer to those of *Malva neglecta* communities than to those of *Hordeum murinum* subsp. *leporinum*. We confirmed the published syntaxon with the rank of *Resedo albae-Glebionenion coronariae* suballiance and its subordination to the *Malvion neglectae* alliance, and we established the type association for this suballiance. *Sisimbrietalia officinalis* J. Tüxen in Lohmeyer et al. 1962 em. Rivas-Martínez, Báscones, T. E. Díaz, Fernández-González & Loidi 1991. *Stellarietea mediae* Tüxen, Lohmeyer & Preising ex von Rochow 1951.

## 1. Introduction

Taxonomy and phytosociology are two long-established tools that underlie the correct interpretation of vegetation and habitats [1,2]. There are numerous studies in these two fields (often as combined works or reviews) that confirm their importance and in which new plant taxa and syntaxa are described (e.g., [3]). Among the many types of vegetation that have been given special importance, we must consider the communities of nitrified grasslands that have been studied by different authors. Gutte [4] proposed the *Malva neglecta* communities in the *Malvenion neglectae* suballiance, and later Rivas-Martínez [5] proposed the *Malvenion parviflorae* suballiance; years later, Rivas-Martínez et al. [6] included the *Urtico urentis-Malvetum neglectae* communities in the suballiance described by Gutte and eight different associations in *Malvenion parviflorae*, subordinating both suballiances to the *Chenopodion muralis* alliance. Both authors carried out exhaustive studies from a phytosociological point of view but did not provide edaphic data. Subsequently, a group of associations of nitrophilic character [7] were studied from the phytosociological and edaphic points of view; among the syntaxa studied, there are the grasslands with *Malva neglecta*, *M. parviflora*, *Chyrsanthemum coronarium* (=*Glebionis coronaria*), and *Hordeum murinum* subsp. *leporinum*. More recently, Cano-Ortiz [8] carried out a study on the *Hordeion leporini* alliance in the Western Mediterranean area, providing relevés from Greece, Italy, Morocco, Spain, and Portugal. In this study, due to the similarity in the ecology and distribution of *Malva neglecta* and *M. parviflora*, the authors state that, taking into account the edaphic parameters and distribution of both *Malva* species and having priority in the name of the suballiance described by Gutte, the suballiance *Malvenion neglectae* should be maintained with its new alliance rank *Malvion neglectae.*

The importance of the communities of *G. coronaria* and *G. discolor* is based on two aspects. Firstly, by acting as indicators of edaphic nutrients and presenting requirements in soil texture and oxidizable organic matter (MOO), and secondly, by acting as CO_2_ sinks and presenting a high biomass. Considering these functions, these communities provide magnificent ecosystem services. Consequently, it is necessary to study the diversity of associations dominated by *Glebionis*, since this is the basis for managers to be able to apply the ecosystem services offered by these communities.

The grassland communities dominated by *G. coronaria* and *G. discolor* were included by several authors [5,9,10,11,12,13,14,15,16,17] in the *Hordeion leporini* alliance, which is composed of grasslands of Mediterranean optimum, subnitrophilous or nitrophilous character, and spring growth. Grasslands that reach sub-Mediterranean and central European territories with sub-nitrophilous communities were included by different authors in alliances such as *Taeniathero-Aegilopion geniculatae* [7,18], while the nitrophilous communities of *Hordeion leporini* and *Malvion neglectae* were studied by Rivas-Martínez [9], and more recently by Cano-Ortiz et al. [8], who proposed the suballiance *Resedo albae-Chrysanthemenion coronariae* Cano Ortiz et al., 2014 [8]. Once the taxonomy of *Glebionis* spp. was established [19,20], as an objective, we set out to adapt the syntaxonomic nomenclature, using the new taxonomy according to the International Code of Phytosociological Nomenclature (ICPN), making the appropriate corrections of names, and incorporating new syntaxa [21].

These ruderal communities have been ignored over time, when in fact they contain great edaphic and thermoclimatic information since *G. coronaria* is always located in thermo-mediterranean environments, not reaching the meso-mediterranean, contrary to what happens with *G. discolor*.

## 2. Results

### 2.1. Cluster Analysis

The cluster analysis for the 81 samples from Spain and Portugal (Figure 1) reveals the difference between 5 plant associations: ArGc (20 relevés) *Anacyclo radiati-Glebionetum coronariae*; KaGc (8 relevés) *Klaseo alcalae-Glebionetum discoloris;* RtGc (8 relevés) *Reichardio tingitanae-Glebionetum coronariae*; AvG (11 relevés) *Anacyclo valentinae-Glebionetum coronariae*; and CbGc (34 relevés) *Centaureo baeticae-Glebionetum discoloris*. These 34 relevés of the cluster CbGc correspond to samples taken by Cano Ortiz [8] (CbGc1 to CbGc21), while the samples from CbGc22 to CbGc34 were obtained by García Fuentes and Cano [22,23], García Fuentes et al. [24], and Rivas Martínez et al. [11]. The five groups are corroborated by phytosociological analysis.

The 101 relevés taken in Italy and Greece are separated in the cluster into three clearly distinct groups: A, B, and C (Figure 2). Group A consists of 4 subgroups: A1 (relevés 1–10), *Malvo parviflorae-Chrysanthemetum coronarii*, published by Brullo et al. [25] for Aspromonte massif (Reggio Calabria, Southern Italy); A2 (relevés 73–77) *Hordeo-Centauretum macracanthae*; A3 (relevés 78–82) *Chrysanthemo-Silybetum mariani*; and A4 (relevés 83–92), *Malvo parviflorae-Chrysanthemetum coronarii*, published by Brullo [26] for Sicily (Italy). The four subgroups are closely related to each other. Group B presents subgroup B1 (relevés 44–62), *Malvo parviflorae-Chrysanthemetum coronarii*, published by Ferro [27] for Sicily, and subgroup B2 published by Ferro [28] for the Aeolian Islands (Sicily, Italy) (relevés 63–72), which is included in the association *Lavatero creticae-Chrysanthemetum coronarii.* Finally, group C is composed of subgroup C1 (relevés 11–43), which is separated from the rest of the groups, thus constituting a new syntaxon, and subgroup C2 (relevés 93–101), which contains the most separated samples from all the others due to the fact that they correspond to Greek territories, far from the others.

### 2.2. Results of the Multivariate Analysis

The statistical analysis according to Cano-Ortiz reveals that the *Glebionis* communities are very close to those of *Malvion*, which is the fundamental reason for including them in this alliance and not in *Hordeion leporini*, as has been the case over time. In addition to presenting a high value as an ecosystem service, they can be used as an indicator of nutrients and as a sink for CO_2_. As can be seen in Figure 3, the plant communities included in *Malvion neglectae* present a variability in their floristic composition, largely explained from the nutritional point of view of the soil. In this case, these communities respond well to an increasing gradient in terms of silt content, CE, pH, and Mg and do not desolubilize in sand-rich soils. In contrast, in the plant communities grown in the *Hordeion leporini* alliance, the edaphic variables that best define these communities are sandy soil texture as well as low pH and CE.

Once the differences between the two main groups of *Hordeion leporini* and *Malvion neglectae* communities were established, the structure of the different communities studied, dominated by *Glebionis coronaria* s.l., was analyzed. The result of the exploratory PCA shows the correlation between the floristic composition of the communities studied in the two countries and their relationship with the different edaphic and bioclimatic parameters. The first two factors explain 58.79% of the variability of the plant communities, and the different relationships between the factors and the samples can be seen in Figure 4.

At first, it can be observed that AdGc is a more or less generalist community, but with a preference for high values of Itc, this bioclimatic parameter provides information about the thermicity of the area where this community develops. The low values of cosine squared (r^2^) in the PCA once rotated (0.07 for F1 and 0.068 for F2) suggest that this is a generalist community in the study area.

For the AvGc community, a high correlation can be observed (r^2^ = 0.104 for F2) with the variables related to humidity. These communities are located in Figure 4, totally opposite to the variables that present an increasing humidity gradient; therefore, they are located in dry and sunny places, in addition to having a certain positive correlation with the apparent density of the soil with the C/N ratio.

The communities called CbGc show high negative correlations with Itc and Tp, as well as correlating positively with soils with high CEC, carbonates, clays, and high pH. This shows the appetence of this floristic combination for calcareous loamy-clay soils in cold places; in fact, they are distributed in the southernmost inland areas of the Iberian Peninsula.

The sampling of CnGc in Italy clearly shows a positive correlation with a gradient of increasing humidity or water content. They are located in rainier and relatively warm places, making the bioclimatic parameters the ones that best explain the floristic composition of this community. A negative correlation can also be observed with edaphic parameters related to high pH values or high carbonate content (r^2^ = −0.47).

KaGc correlates positively with the variable C/N and negatively with Ic, and the continentality index provides information about the annual thermal oscillation. This community also correlates significantly with low N contents, compared to other communities, as well as with low moisture or water content indices.

Finally, the RtGc community presents similar affinities and correlations to KaGc in the same variables, although with slightly higher squared cosines; therefore, these variables have greater relevance in the distribution of this plant community.

### 2.3. Canonical Correspondence Analysis Results

Once the phytosociological importance value of each of the species forming the different phytochorions under study was calculated, those taxa whose modified IVI was included in the top 5% (5% percentile) were selected. With these species selected according to their importance within the community, a CCA was carried out with the aim of observing the possible correlations between environmental variables and the presence and abundance of these species. The first two factors account for 39.84% of the variability. Although it is true that this explained variance is a priori low, it should be noted that the CCA seems to be sensitive to the relationship between the number of species and the number of samplings, so much so that in the different simulations and model adjustments carried out, when the number of species to be analyzed was reduced, the explained variance increased because all the variance was explained by a few species. However, it was decided to observe the behavior of the 62 species with the highest IVI, since the bioindicator character of the species is largely preserved.

In principle, differences can be observed in the inclination of various species depending on the country. As can be seen in Figure 5, the CCA clearly shows 5 groups of species. On the one hand, there are species with an inclination for carbonate-rich soils, high pH, and clay soils that are negatively correlated with high Tp, PE, and Iar values, such as *Centaurea baetica*, *Sinapis mairei*, *Diplotaxis catholica,* or *Phalaris minor*. These species characterize the so-called CbGc communities. 

Another group of species has an inclination for clay soils that are poor in sand, rich in carbonates, have a high bulk density, high C/N ratios, and are negatively correlated with variables such as Ic, PEs, and Ios2. These species are *Klasea alcalae*, *Hyparrhenia sinaica*, *Centaurea pullata*, *Crepis vesicaria*, and *Asteriscus acuaticus*. These species characterize the phytochorions called KaGc.

On the other hand, there is another cluster of species strongly correlated with the variables associated with high temperatures (Itc) and negatively correlated with Io and Ioe. Partial negative correlations can also be seen with other variables related to soil carbonate content and high pH. Species such as *Reichardia tingitana*, *Brassica repanda*, and *Tetragonia tetragonioides* are characteristic of RtGc communities. These three species clusters correspond to plant communities present in the Iberian Peninsula, which are well characterized from the point of view of species co-occurrence (floristic composition) as well as from a bioclimatic and edaphic point of view. Another grouping of species responds, with positive correlations, to the variable continentality (Ic), as well as being partially correlated with the variables related to water availability both at annual (Io) and summer (Ioe) levels. From an edaphic point of view, they are negatively correlated with variables such as the C/N ratio (CN) or soil bulk density. Species such as *Astragalus drupaceus* and *Sinapis dissecta* define these communities. These phytochorions correspond to the so-called AdGc, distributed in the central-eastern Mediterranean belt in Greece, being the eastern-most community studied in this manuscript.

Finally, in the analysis of canonical correspondences, a well-defined cluster of plant species with a clear appetence for arid or low humidity climates and high temperatures can be observed. They are positively correlated with variables such as the aridity index (Iar), the positive annual temperature (Tp), or summer evapotranspiration (PEs). From the edaphic point of view, they correlate negatively with variables such as clay content, pH, or carbonate content. Species such as *Anacyclus valentinus*, *Carrichtera annua*, and *Lophocloa pumila* are examples. These phytochorions, or plant co-communities, correspond to AvGc in this manuscript and are located in the most arid areas of Western Europe.

Within this last cluster, we can distinguish a subcluster of species that are not very correlated with edaphic and bioclimatic variables in general, although with an inclination for warm and relatively arid climates, showing a certain correlation with variables such as PE, Iar, or Tp. From the edaphic point of view, they seem to exclude soils rich in carbonates, with a clayey texture, or with a high pH. The species with these appetences are characteristic of the communities sampled in Italy, maintaining a certain relationship with AvGc. Species such as *Centaurea napifolia*, *Calendula fulgida*, *Lotus ornithopodioides*, and *Galactites elegans* (synonym of *Galactites tomentosus* Moench) respond to these characteristics in the *Glebionis coronaria* communities in Italy. These phytochorions have been referred to as CnGc.

## 3. Discussion

The communities dominated by *Glebionis coronaria*, *G. discolor,* and *Hordeum leporinum* have structural, edaphic, and floristic differences. According to Cano-Ortiz et al. [8], at the edaphic level, *Glebionis* communities present soil parameter values of MO, Nt, P, K, and Mg that are closer to those of *Malvenion neglectae* Gutte 1966 communities than to those of *Hordeion leporinii*. The frequent presence of *Malvenion neglectae* species in *Glebionis* communities and the structure of *Malvenion neglectae* [4] were reason enough for Cano-Ortiz et al. [8] to propose the new suballiance *Resedo albae-Chrysanthemenion coronarii* and subordinate it to *Malvion neglectae* (Gutte 1966) Cano Ortiz et al. 2014, whose name we now correct according to the newly established taxonomy *Resedo albae-Glebionenion coronariae* (Gutte 1966) Cano Ortiz et al. 2014 *corr.* (Table 1).

All the communities of the genus *Glebionis* are of great interest for land management, so it has been necessary to describe them phytosociologically. These plant associations have a narrow edaphic ecology in that they are bioindicators of edaphic nutrients [8], and it is necessary for managers to know the associations described with the abundance of the species. The communities dominated by *G. coronaria* for the thermo-Mediterranean environments of southwestern Iberia (Spain and Portugal) were published by Cano-Ortiz et al. [7] with the name *Anacyclo radiati-Chrysanthemetum coronarii* (Rivas-Martínez 1978) Cano-Ortiz et al. 2009, making it necessary to correct the name in all syntaxa (ICPN article 44) *Anacyclo radiati-Glebionetum coronariae* (Rivas-Martínez 1978) Cano-Ortiz et al. 2009 corr., since this association included the suballiance *Resedo albae-Chrysanthemenion coronarii* Cano Ortiz et al. 2014 and the alliance *Malvion neglectae* (Gutte 1966), whose name we now correct according to the newly established taxonomy *Resedo albae-Glebionenion coronariae* (Gutte 1966) Cano Ortiz et al. 2014 corr. Because Cano-Ortiz et al. [8] do not give the type for the suballiance, it is typified in this work, and we propose as typus *Resedo albae-Glebionetum coronariae* O. Bolòs & Molinier 1958 nom. corr. Indeed, due to the doubts raised about the taxonomy of *G. coronaria*, Cano et al. [19] carried out a taxonomic study on both *Chrysanthemum coronarium* var. *concolor* and var. *discolor* and distinguished the taxa of *G. coronaria* and *G. discolor*. Based on this new taxonomy, we establish a new syntaxonomy for the *Glebionis s.l.* communities being included in the suballiance *Resedo albae-Glebionenion coronariae* (Cano-Ortiz et al., 2014) *nom. corr.,* which is subordinate to *Malvion neglectae.* However, these authors do not propose the type of the suballiance, so we designate as type the association *Resedo albae-Glebionetum coronariae* O. Bolòs & Molinier 1958 *nom. corr.*

Based on the results obtained for Spain and Portugal for the association *Anacyclo radiati-Glebionetum coronariae*, we propose the following new associations:*Centaureo baeticae-Glebionetum discoloris* Cano-Ortiz *ass. nova* (Table 2, Relevés 1–21, *holotypus* rel. 6), an upper thermo-mediterranean and lower meso-mediterranean grassland growing on nitrified basic substrates in the Betic territories, characterized by *G. discolor* and *Centaurea pullata* subsp. *baetica* differential species compared to *Resedo albae-Chrysanthemetum coronarii*, described by Bolòs and Molinier [29,30] for the thermo-mediterranean areas of Mallorca, and extending its area to the thermo-mediterranean areas of Valencia [31]. Three types of grasslands dominated by *G. coronaria* develop in southeastern Iberian thermo-mediterranean territories on basic and neutral substrates.*Klaseo alcalae-Glebionetum discoloris ass. nova* (Table 3, Relevés 1–8, *holotypus* rel. 8), a very frequent plant community in the Malacitano-Almijarense biogeographic sector (Malaga province, Spain), which develops on nitrified neutro-basic substrates of anthropized areas such as roadsides and abandoned places. Its dominant species are *G. coronaria*, *Klasea alcalae,* and *Sinapis alba* subsp. *mairei*.In the Alpujarreño-Gadorense biogeographic unit (Granada province, Spain), the *Glebionis coronaria*-dominated grassland continues to prevail, with a similar ecology to the previous one but with a different floristic composition: *Reichardio tingitanae-Glebionetum coronariae ass. nova* (Table 4, Relevés 1–9, *holotypus* rel. 4).Finally, in the semi-arid thermo-mediterranean territories on basic substrates rich in organic matter of the Almerian sector, there is a community of *G. coronaria* differentiated from the previous ones by the presence of *Anacyclus valentinus Beta vulgaris* and *Carrichtera annua* among other species, which allows us to propose the association *Anacyclo valentinae-Glebionetum coronariae ass. nova* (Table 5, Relevés 1–11, *holotypus* rel. 2).

In previous research by Cano et al. [19], we established the species *G. coronaria* (L.) Cass. ex Spach for exclusively thermo-mediterranean environments and *G. discolor* (d’Urv.) Cano, Musarella, Cano-Ortiz, Piñar Fuentes, Spampinato & Pinto Gomes for thermo- and meso-mediterranean environments. The morphometric study confirmed different dimensions of achene wings and different arrangements of their glands (Figure 6).

This work was carried out due to several previous taxonomic errors that have remained uncorrected since the time of Linnaeus. Unfortunately, attempts at lectotypification of *Chrysanthemum coronarium* L. by Dillon were made erroneously. However, Turland [30] confirmed two varieties and proposed a new combination under the name *Glebionis coronaria* var. *discolor* (d’Urv.) Turland, using *Chrysanthemum coronarium* var. *discolor* d’Urv. as a basionym. After the morphometric study and bioclimatic distribution, different authors have reported both taxa with specific ranks: Cueto et al. [32] for Spain and Bertolucci et al. [33,34] for Italy.

The morphometric differences between *G. coronaria* and *G. discolor* are supported by the phytochemical study of Ivashchenko et al. [35], which concludes that there are differences in the amounts of carotene, vitamins, and other molecules present in both species due to the influence of environmental factors. Recently, Gallucci et al. [20] carried out a study of the genetic diversity between *G. coronaria* and *G. discolor* through AFLP markers, using material from Spain, Italy, and Portugal, and reached the conclusion that, using the mentioned markers, there are genetic differences between both species.

Recently, both *G. coronaria* and *G. discolor* have been accepted in the Flora Vascular de Andalucía (https://www.florandalucia.es/index.php/glebionis-coronaria#:~:text=Antimonia%2C%20belide%20menuda%2C%20besantemon%20oloroso,mogigato%2C%20mohinos%2C%20mohinos%20bastos%2C accessed on 12 February 2024), as well as *G. coronaria* in the Flora Canaria. However, Benedi [36] includes *G. discolor* within *G. coronaria* (L.) Spach, and in remarks he says that several authors consider with distinct rank two forms, those with totally yellow ligule and those with white ligule with a yellow base. It is not surprising that he includes *G. discolor* in *G. coronaria*, since his study predates the molecular study. 

Taking into consideration articles 40–45 of the ICPN [21], it is not possible to maintain the syntaxonomy that is based on the species *C. coronarium*, when a vast majority of researchers support *G. coronaria* and *G. discolor*. For these reasons, we propose the correction of the names.

Although Álvarez de la Campa [37] described the association *Asphodelo fistulosi-Hordeetum leporini* A. & O.Bolòs in O.Bolòs 1956 and the sub-association *chrysanthemetosum coronarii* Álvarez de Campos 2003, the only differential being the taxon *Chrysanthemum coronarium* L. [=*Glebionis coronaria* (L.) Cass. ex Spach], due to its thermo-mediterranean environment and similar floristic composition, it could be assimilated to the new AvG association. All these associations are distributed in the south and east parts of the Iberian Peninsula (Figure 7).

In Italian territories, several authors have followed the Spanish and Portuguese syntaxonomy with respect to nitrophilous and sub-nitrophilous communities [25,26,27,28,39,40,41,42,43,44,45,46]. For Italy and Greece, Ferro [27] described the association *Malvo parviflorae-Glebionetum coronariae* Ferro 1980 corr. for Sicily (B1), which is corroborated by Brullo [26] in Sicily, together with the new associations described by this author, *Hordeo-Centauretum macracanthae* (A2) and *Glebiono-Silybetum mariani* Brullo 1983 corr. (A3). However, Brullo et al. [25] studied these *Glebionis* communities in southern Italy and included them in *Malvo parviflorae-Glebionetum coronariae*. Community (A1) in the cluster (Figure 2) is very far from the one described by Ferro, being closer to (A2), (A3), and (A4), so these grasslands should be included in the syntaxa described by Brullo [26] (Figure 8).

Group B is constituted by relevés published by Ferro [27,28] for Sicily and for the Aeolian Islands, broken down in the cluster into two subgroups: (B1) *Malvo parviflorae-Glebionetum coronariae* and (B2) *Lavatero creticae-Glebionetum coronariae* Ferro 2004 corr., described for the Aeolian Islands. Both associations have strong floristic differences with respect to the group (C1) of the cluster, which allows us to propose a new association (Relevés 1–33) for the territories of Reggio Calabria, characterized by species absent in the association *Lavatero creticae-Glebionetum coronariae* (B2), these being the differences of the new association *Centaureo napifoliae-Glebionetum coronariae ass. nova* (Table 6, Relevés 1–19, *holotypus* rel. 16), an association differentiated from *Centauretum napifoliae* Brullo 1983. This association grows in dry environments on basic substrates in the thermo-mediterranean belt. The subgroup (B2) *Malvo parviflorae-Glebionetum coronariae* published by Brullo [26] for Sicily differs from the same (A1) communities published by Brullo et al. [25] for Reggio Calabria. The group (C1) based on relevés from Sicily and Reggio Calabria belongs to a new syntaxon, floristically differentiated from the rest of the relevés of C1 by the presence, among other species, of *Calendula suffruticosa* subsp. *fulgida*, *C. suffruticosa* subsp. *fulgida x Calendula arvensis*, and *Valeriana graciliflora* (=*Fedia graciliflora*). This community is present on basic and loamy substrates in dry-subhumid environments of the lower thermo- and meso-mediterranean territories, which allows us to propose the new subassociation *Centaureo napifoliae-Glebionetum coronariae calenduletosum fulgidae subass. nova* (Table 6, Relevés 20–33, *holotypus* rel. 31), while the grasslands dominated by *G. coronaria* in Greece belong to subgroup (C2), which establishes a new syntaxon named *Astragalo drupacei-Glebionetum coronariae ass. nova* (Table 7, Relevés 1–9, *holotypus* rel. 2), an association located on thermo-mediterranean basic substrates (Figure 9).

Proposed syntaxonomical scheme

*Stellarietea mediae* Tüxen, Lohmeyer & Preising ex von Rochow 1951

*Chenopodio-Stellarienea* Rivas Goday 1956

*Sisimbrietalia officinalis* J. Tüxen in Lohmeyer et al. 1962 *em.* Rivas-Martínez, Báscones, T. E. Díaz, Fernández-González & Loidi 1991

*Malvion neglectae* (Gutte 1966) Cano Ortiz, Biondi, Pinto-Gomes, Del Río & Cano 2014 

*Resedo albae-Glebionenion coronariae* Cano-Ortiz, Biondi & Cano ex Cano Ortiz, Biondi, Pinto-Gomes, Del Río & Cano 2014 *corr*.

*Resedo albae-Glebionetum coronariae* O. Bolòs & Molinier 1958 *nom. corr.*

*Anacyclo radiati-Glebionetum coronariae* (Rivas-Martínez 1978) Cano-Ortiz et al. *nom. corr.*

*Malvo parviflorae-Glebionetum coronariae* Ferro 1980 *nom. corr.*

*Lavatero creticae-Glebionetum coronariae* Ferro 2004 *nom. corr.*

*Glebiono-Silybetum mariani* Brullo 1983 *nom. corr.*

*Reichardio tingitanae-Glebionetum coronariae ass. nova* (Spain) 

*Anacyclo valentine-Glebionetum coronariae ass. nova* (Spain)

*Centaureo baeticae-Glebionetum discoloris ass. nova* (Spain)

*Klaseo alcalae-Glebionetum coronariae ass. nova* (Spain)

*Astragalo drupacei-Glebionetum coronariae ass. nova* (Greece)

*Centaureo napifoliae-Glebionetum coronariae ass. nova* (Italy)

*Centaureo napifoliae-Glebionetum coronariae calenduletosum fulgidae subass. nova* (Italy)

## 4. Materials and Methods

### 4.1. Characterization of the Territory and Study Area

The study territory corresponds to the western Mediterranean (Spain, Italy, Portugal, and Greece). Samples were taken in the spring for several years. The study was initiated by Cano-Ortiz, who obtained edaphic and phytosociological samples in Spain and Italy, and continued later [7,8,18]. In order to differentiate the communities dominated by *G. coronaria* and *G. discolor* in the European Mediterranean, we rely on 101 phytosociological relevés for Italy and Greece (51 for Italy from other authors, while the 41 from Italy and the 9 from Greece are their own samples) carried out by us and by other authors (Table 8 and Table 9) and 81 new samples for Spain and Portugal. 

For the ecological characterization of the different communities studied, edaphic and climatic data have been used. On the one hand, the edaphic data were obtained from the Maps of Soil Chemical Properties at European Scale based on LUCAS 2009/2012 topsoil data [48] and Topsoil Physical Properties for Europe (based on LUCAS topsoil data) [49]. The climate data were obtained from monthly summaries of average maximum and minimum temperatures and precipitations from WorldClim [50] raster layers of 2.5′ arc resolution from 1960 to 2021. The different monthly averages have been produced using map algebra in QGIS 3.22 software. Subsequently, the different bioclimatic indices have been calculated according to the Rivas-Martínez World Bioclimatic Classification [51]. The different variables for each sampling are shown in Table 10.

### 4.2. Statistical Analysis

Cluster statistical analysis was applied to differentiate the groups of plant communities. This analysis was corroborated and adjusted with the phytosociological study through the presence/absence of species, biogeographical distribution, and ecology. For the statistical ordination treatment, the computer package Community Analysis Pakge III was used. To establish the distribution of the two types of *G. coronaria* and *G. discolor* communities, we used the bioclimatic map established by Cano et al. [19] (Figure 10) and our own phytosociological sampling.

Multivariate analyses were used for the ecological characterization of the different communities studied. Taking the edaphic and bioclimatic data as variables, a principal components analysis was carried out, previously selecting the variables that best explained the variability of the communities, as well as a canonical correspondence analysis with the aim of relating the co-occurrence of the different species with the different edaphic and bioclimatic variables. The importance of each of the bioclimatic and edaphic variables was tested using the Kaiser-Meyer-Olkin (KMO) test. This index provides information on when the data are suitable for factor analysis. It is used to assess whether the relationship between variables is strong enough for the factor analysis to produce significant and reliable results. The cut-off criterion was to choose variables with a KMO index > 0.5 (Table 10). The PCA was performed using Pearson’s correlation and with an Oblimin-type rotation of the axes. The oblimin rotation seeks to minimize the number of variables that have high loadings on more than one factor. This can be useful when factors are expected to be correlated in reality, such as in situations where the underlying variables share some relationship or overlap conceptually.

With the environmental variables selected on the basis of the factor analysis, the modified phytosociological importance index (IVI) was then calculated for each of the plant species present in the different samples. Based on the relative frequency in each sampling, the relative dominance in each sampling, and the inverse of the relative occurrence in the different clusters determined in the previous ordination analysis, this step was carried out to “penalize” the importance of the more euryoecious species in favor of the more stenoecious ones.
$$IVI=\frac{\left(Fr+Dr\right)}{Frt}$$
where Fr is the relative frequency of occurrence of the species in each sampling cluster; Dr is the mean relative dominance (measured in cover) in each sampling cluster; and Frt is the relative frequency of occurrence in all sampling clusters. Frt penalizes those species found in many different communities.

Subsequently, based on the preliminary results of the factorial analysis and PCA, the analysis of the main components, and the phytosociological importance of each species, a canonical correspondence analysis (CCA) was carried out with the aim of observing ecological patterns according to each of the distribution gradients of the environmental variables in the composition of each phytocorion or plant community studied.

In order to compare the different communities with each other according to soil composition and bioclimate, comparative analyses of variance were carried out. Previously, an exploratory analysis of the data was carried out to check the distribution, normality, and homoscedasticity of the data for a better choice of comparative methods. For this purpose, the Shapiro–Wilks test was used to check the normality of the data. This test showed that most of the variable distributions did not follow a normal distribution, so non-parametric methods were chosen. For the comparison between the different plant communities studied, the Kruskal–Wallis method of comparisons between medians was used, a non-parametric method analogous to the ANOVA analysis.

## 5. Conclusions

This study clearly highlights the differentiation of the Italian grasslands from the Iberian ones, which are separated into two alliances: the *Hordeion leporini* alliance distributed throughout the western Mediterranean basin, from which we segregate the communities of *G. coronaria* and *G. discolor* according to their different edaphic, floristic, and structural characteristics that allow us to include these grasslands in the *Resedo albae-Glebionenion coronariae* sub-alliance, which we include in the *Malvion neglectae* alliance. According to the new taxonomy of the genus *Glebionis* [19,20], name corrections are made according to the Code of Phytosociological Nomenclature. The statistical study of 81 samples for Spain and Portugal and 101 for Italy and Greece allowed us to establish seven new syntaxa, six of which have association rank and one with subassociation rank.

## Figures and Tables

**Figure 1 plants-13-00568-f001:**
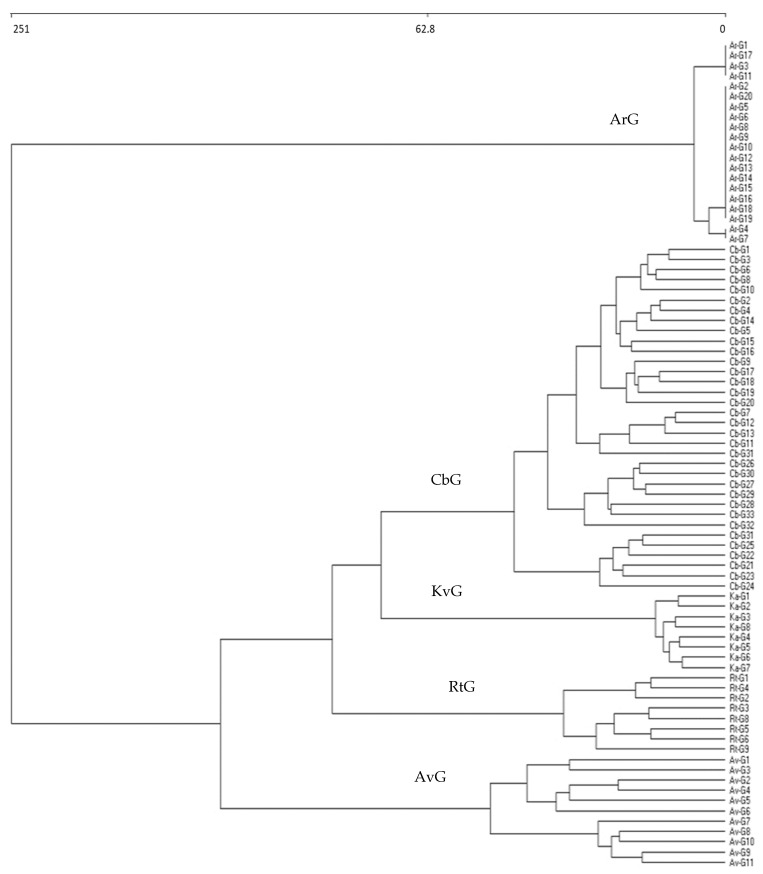
Cluster analysis of grasslands dominated by *Glebionis coronaria* and *Glebionis discolor* in the Iberian Peninsula.

**Figure 2 plants-13-00568-f002:**
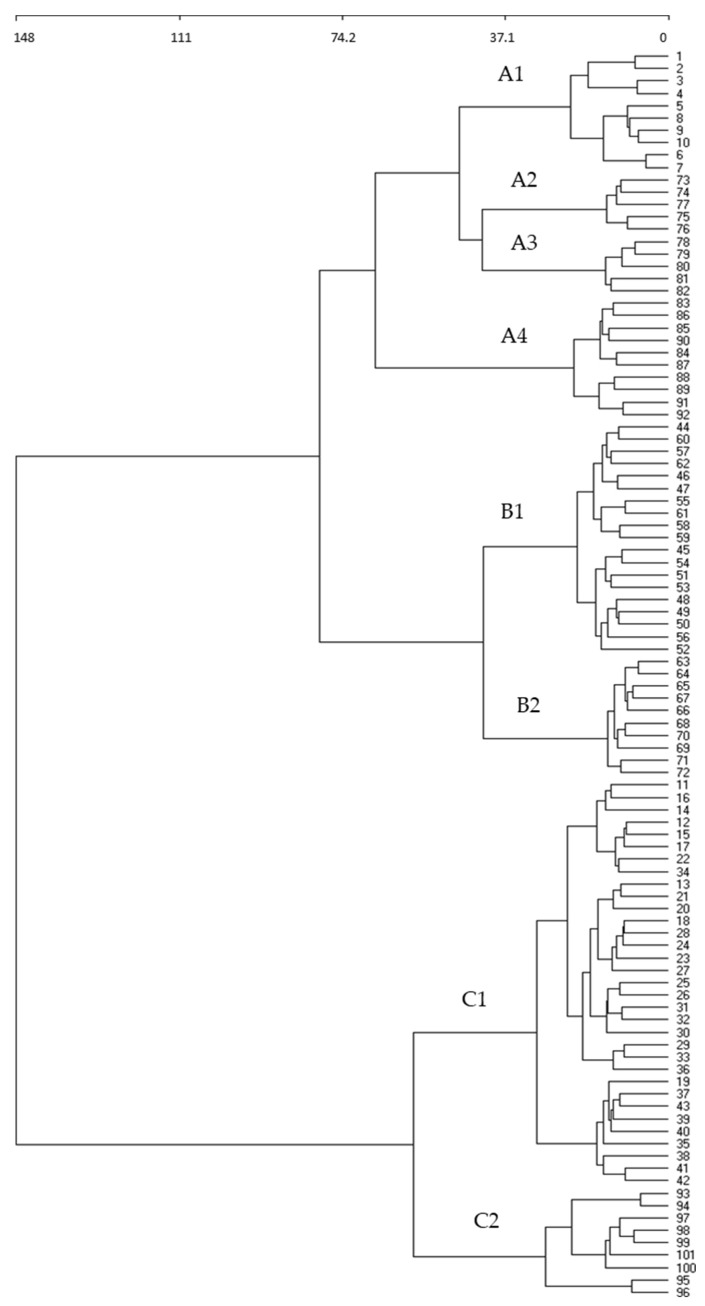
Cluster analysis of grasslands dominated by *Glebionis coronaria* and *Glebionis discolor* in Italy and Greece.

**Figure 3 plants-13-00568-f003:**
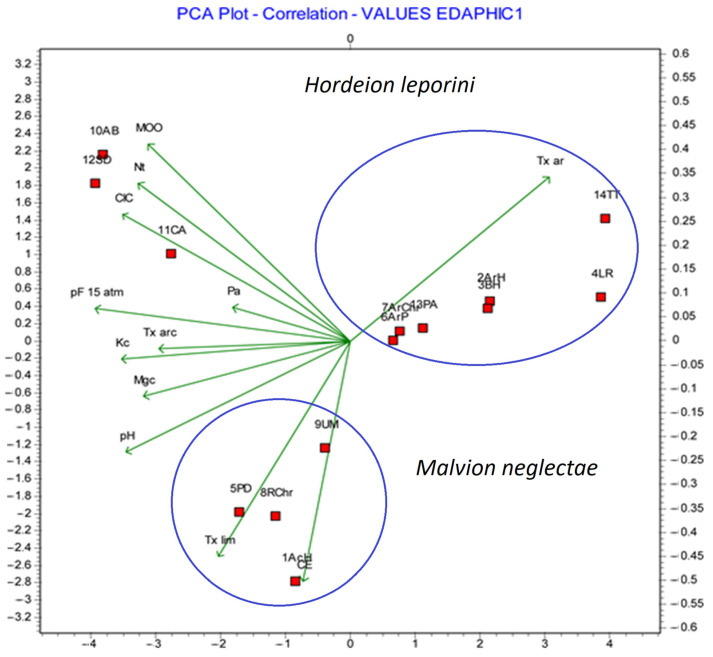
Edaphic relationships of *Hordeion leporini* and *Malvion neglectae* communities [8].

**Figure 4 plants-13-00568-f004:**
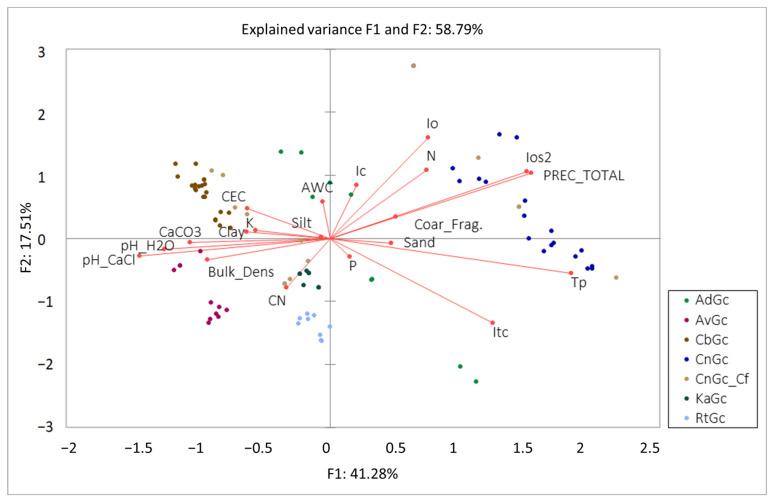
Representative biplot of the first two factors (F1 and F2) of the principal component analysis (PCA) of the communities studied in Spain, Italy, and Greece. AdGc = *Astragalo drupacei-Glebionetum coronariae* (Greece); AvGc = *Anacyclo valentinae-Glebionetum coronariae* (Spain); CbGc = *Centaureo baeticae-Glebionetum discoloris* (Spain); CnGc = *Centaureo napifoliae-Glebionetum coronariae typicum* (Italy); CnGc_Cf = *Centaureo napifoliae-Glebionetum coronariae calenduletosum fulgidae* (Italy); KaGc = *Klaseo alcalae-Glebionetum discoloris* (Spain); RtGc = *Reichardio tingitanei-Glebionetum coronariae* (Spain).

**Figure 5 plants-13-00568-f005:**
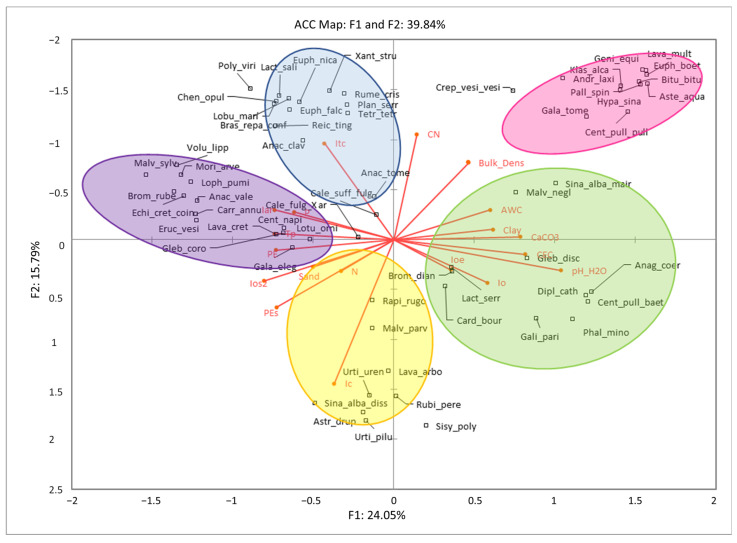
CCA correlation analysis between bioclimatic and edaphic parameters of the abundance of species that make up the different communities.

**Figure 6 plants-13-00568-f006:**
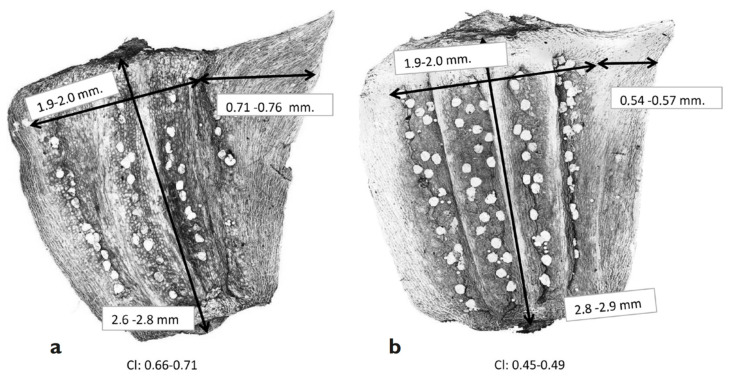
Disc cypsela of *Glebionis coronaria* (**a**) and *G. discolor* (**b**) photographed with high-resolution confocal microscopy [19].

**Figure 7 plants-13-00568-f007:**
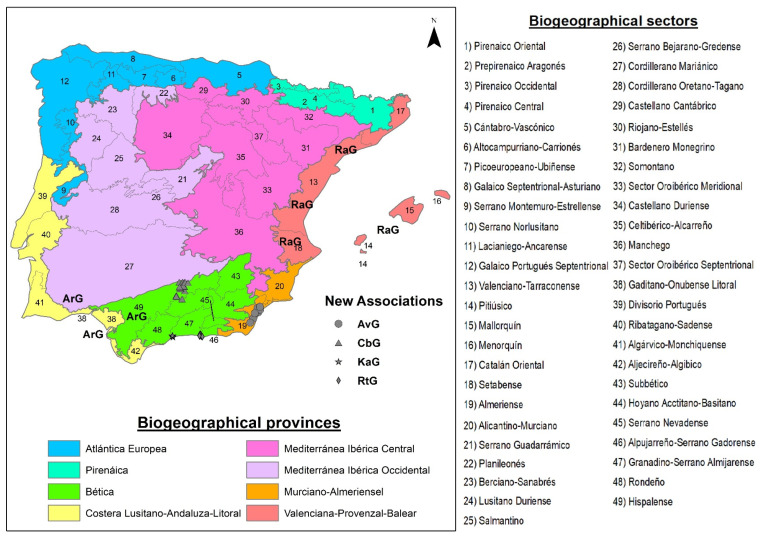
Distribution map of associations in the Iberian Peninsula. Map adapted from Rivas-Martínez et al. [38].

**Figure 8 plants-13-00568-f008:**
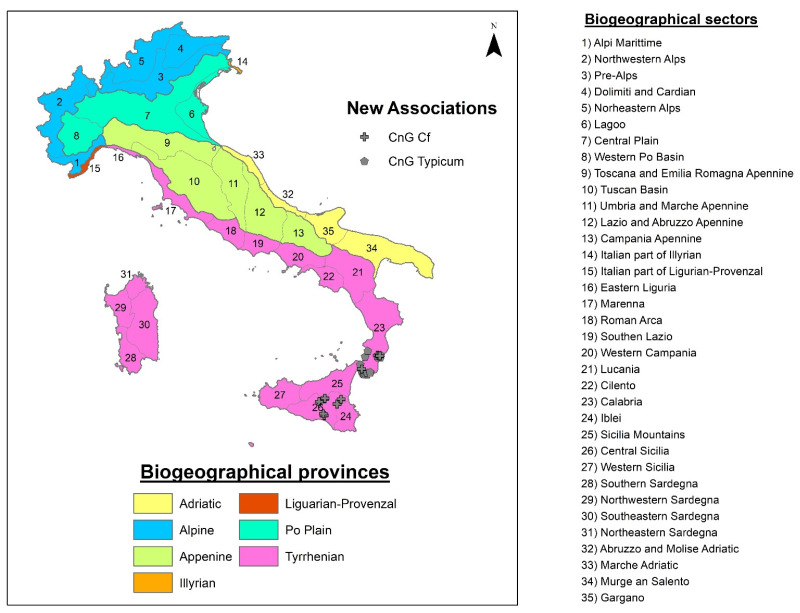
Distribution map of associations in Italy. Map adapted from Blasi et al. [47].

**Figure 9 plants-13-00568-f009:**
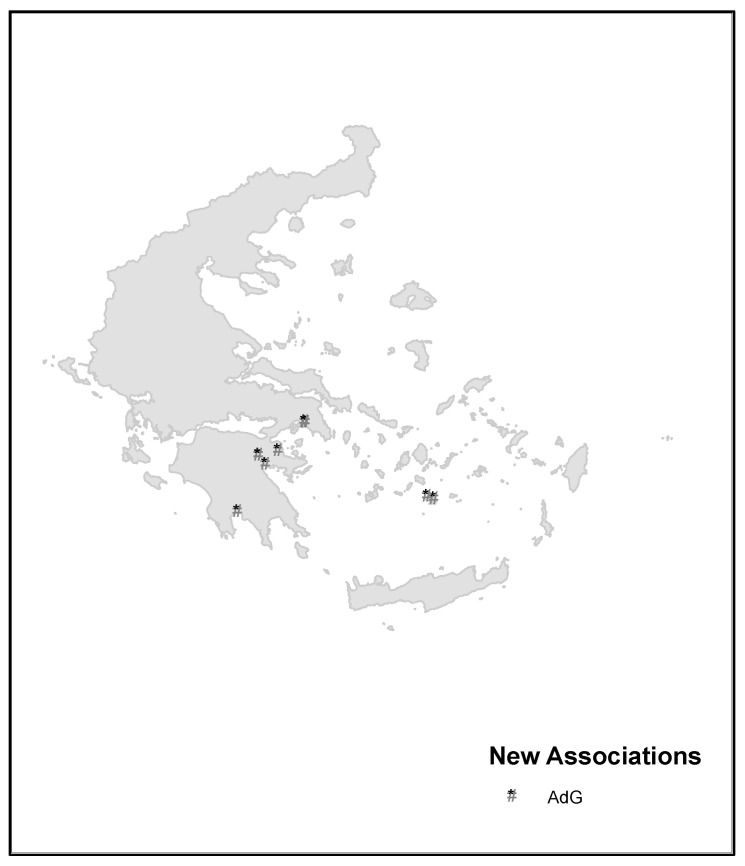
Distribution map of associations in Greece.

**Figure 10 plants-13-00568-f010:**
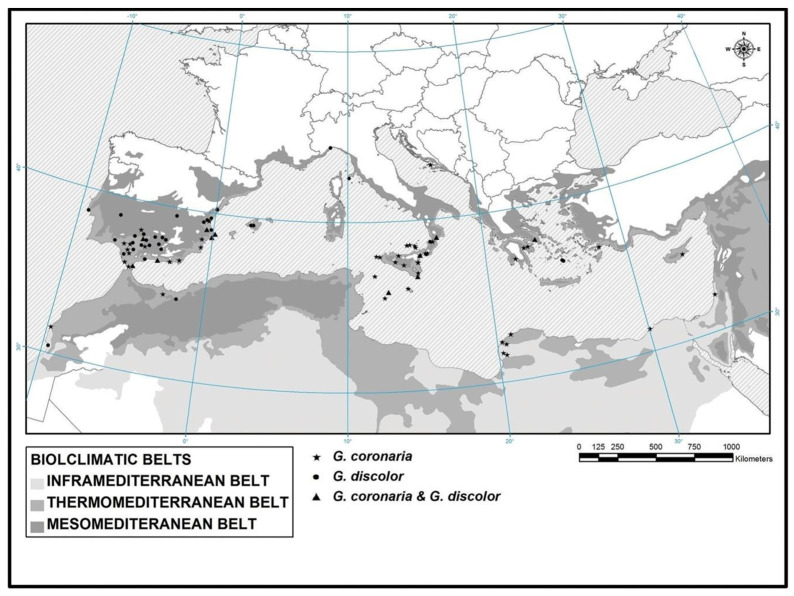
Bioclimatic distribution of *Glebionis coronaria* and *G. discolor* in the Mediterranean basin [19].

**Table 1 plants-13-00568-t001:** Edaphic analysis of the communities dominated by *Hordeum leporinum*, *Glebionis coronaria,* and *Glebionis discolor* [8].

	AcH	ArH	BsH	ArC	CbG	UuM
CIC	15.365	9.131	10.544	12.328	11.68	10.889
MOO	1.541	1.56	1.667	1.622	1.574	1.904
Nt	0.115	0.102	0.133	0.105	0.129	0.179
Pa	9.789	13.957	15.4	26.9	20.95	36.19
Mgc	1.683	1.856	1.068	2.131	2.716	1.698
Kc	0.79	0.256	0.375	0.698	1.476	1.266
pF 15 atm	15.322	8.613	8.203	11.975	14.24	13.197
Tx arc	17.758	19.78	14.503	19.763	24.24	21.293
Tx ar	20.448	62.411	54.254	55.826	37.85	46.001
Tx lim	61.794	17.803	31.245	24.413	37.94	32.712
CE	0.355	0.209	0.122	0.193	0.491	0.565
pH	8.275	7.43	7.475	7.77	7.943	7.776
pF^15atm^	15.322	8.613	8.203	11.975	14.238	13.197

CIC = cation exchange capacity in meq/100 g; MOO = oxidable organic matter in %; Nt = total nitrogen in %; Pa = assimilable phosphorus in ppm; Mgc = exchangeable magnesium in meq/100g; Kc = exchangeable potassium in meq/100 g; pF 15 atm = pressure at 15 atm (water retention capacity) in %; Tx arc = clayey texture in %; Tx ar = sandy texture in %; Tx lim = silty texture in %; EC = conductivity mmhos/cm; pH. AcH—*Anacyclo clavati-Hordeetum leporini* Cano-Ortiz et al. 2009. ArH—*Anacyclo radiati-Hordeetum leporinii* O. Bolòs & Rivas-Martínez in Rivas-Martínez 1978. BsH—*Bromo scoparii-Hordeetum leporinii* Rivas Martínez 1978. ArGb—*Anacyclo radiati-Glebionetum coronariae* (Rivas-Martínez 1978) Cano-Ortiz et al., 2009 *corr.* CbG—*Centaureo baeticae-Glebionetum coronariae ass. nova*. UuM—*Urtico urentis-Malvetum neglectae* (Knapp) Lohmeyer in Tüxen 1950.

**Table 2 plants-13-00568-t002:** *Centaureo baeticae-Glebionetum discoloris ass. nova* (Andalusia, Spain. Inv. 1–21, **holotypus** inv. 6).

N° inventory	1	2	3	4	5	**6**	7	8	9	10	11	12	13	14	15	16	17	18	19	20	21
N° cluster	CbG1	CbG2	CbG3	CbG4	CbG5	**CbG6**	CbG7	CbG8	CbG9	CbG10	CbG11	CbG12	CbG13	CbG14	CbG15	CbG16	CbG17	CbG18	CbG19	CbG20	CbG21
Surface (m^2^)	1	1	1	1	1	**1**	1	1	1	1	1	1	1	1	1	1	1	1	1	1	1
Coverage (%)	100	95	100	100	90	**100**	100	100	100	100	100	100	100	100	100	100	100	100	100	100	100
Altitude (m)	240	298	230	407	235	**412**	369	373	438	406	319	207	179	200	196	217	446	446	333	249	240
Orientation	.	SW	S	.	E	**.**	.	.	.	.	.	.	.	.	.	.	.	.	.	.	.
Inclination (%)	.	10	2	.	1	**.**	.	.	.	.	.	.	.	.	.	.	.	.	.	.	.
Average hight veg (cm)	130	110	100	95	80	**150**	170	145	130	160	140	160	160	160	120	160	160	200	135	130	200
**Characteristics association and upper units**																					
*Glebionis discolor* (d’Urv) Cano et al.	5	5	5	5	4	**5**	5	5	5	5	5	5	5	5	5	5	5	5	5	5	5
*Centaurea pullata* L. subsp. *baetica* Talavera	+	1	+		+	**1**	1	+	+	+	+		1	1		+		1	1	+	+
*Bromus diandrus* Roth s.l.	2	2	+	2		**1**	1	1	1	1	1	1	+		+	+	+				
*Erodium malacoides* (L.) L’Hér.	1	1	+	1	2	**1**	1	+		1		1	1	2		+					
*Avena sterilis* L.	2		1			**1**	2	1		1	1	2	+		2	2	+	2			
*Lactuca serriola* L.		+		+	1	**1**		+		+			1	+		+	+	+			+
*Malva parviflora* L.						**+**	1		1		1	+	1	+		1					+
*Malva neglecta* Wallr.	+	2	+	+	+									1	+	1					+
*Calendula arvensis* L.	1			1	3	**1**				+	1				+	+				+	
*Sinapis alba* L. subsp. *mairei* (H. Lindb.) Maire	1					**2**	+		+		+	1						+	+		
*Medicago polymorpha* L.			+	+					3			+	+	1	1		+				
*Echium plantagineum* L.							+		+	3	+	+	1							1	
*Rapistrum rugosum* (L.) All.							2			+	+	+	2								+
*Hordeum murinum* L. subsp. *leporinum* Link		2	1	2					2						+	+					
*Diplotaxis catholica* (L.) DC.				+	+						1				1			+			+
*Galium parisiense* L.										+	+		1			+	+				
*Rumex conglomeratus* Murray							1	+					+						+		
*Papaver rhoeas* L.						**+**					1				2		+				
*Anagallis coerulea* Schreber		+									+	+					+				
*Sonchus tenerrimus* L.						**+**		+			+										
*Lolium rigidum* Gaudin									+						2			+			
*Eruca vesicaria* (L.) Cav.		+								1								+			
*Crepis vesicaria* L. subsp. *haenseleri* (Boiss.) Sell.												+							+	1	
*Avena longiglumis* Durieu																			1	+	+
*Nonea vesicaria* (L.) Rchb.					+					+											
*Melilotus indica* (L.) All.																				+	+
*Medicago arabica* (L.) Hudson		2			+																
*Lolium multiflorum* Lam.									+	+											
*Lathyrus clymenum* L.										1									+		
*Geranium rotundifolium* L.				+																+	
*Erodium moschatum* (L.) L’Hér				1											+						
*Erodium ciconium* (L.) L’Hér																			1		+
*Diplotaxis virgata* (Cav.) DC.	+		+																		
*Bromus madritensis* L.															+				+		
*Vicia sativa* L. subsp. *nigra* (L.) Ehrh.												+									
*Vicia monthana* Retz.,																					+
*Vicia hybrida* L.																1					
*Vicia benghalensis* L.									2												
*Stellaria alsine* Grimm					1																
*Stachys arvensis* (L.) L.											+										
*Sonchus oleraceus* L.					1																
*Raphanus raphanistrum* L.											+										
*Plantago lanceolata* L.											+										
*Plantago lagopus* L.					2																
*Ononis viscosa* L.																				+	
*Ononis biflora* Desf.																				+	
*Medicago orbicularis* (L.) Bartal.									2												
*Medicago doliata* Carming. var. *muricata* Heyn.												+									
*Lolium temulentum* L.													1								
*Lactuca virosa* L.					1																
*Glossopappus macrotus* (Durieu) Briq.						**+**															
*Gastridium ventricosum* (Gouan) Sch. et Th.															+						
*Galactites tomentosa* Moench						**+**															
*Fumaria reuteri* Boiss.															+						
*Euphorbia helioscopia* L.				+																	
*Bromus intermedius* Guss.																+					
*Brassica nigra* (L.) Koch	1																				
*Bellardia trixago* (L.) All.																				+	
*Avena barbata* Potter														+							
**Companion species**																					
*Silybum marianum* (L.) Gaertner			1				1		+			+	+	1	+		1	+	+		+
*Carduus bourgaeanus* Boiss. & Reuter					1		+	1			+	+	2	1	+	1					1
*Foeniculum vulgare* Miller subsp. *piperitum* (C. Presl) Bég	+		+			**+**		+		+										+	
*Phalaris minor* Retz.								1							1	1		+			+
*Convolvulus althaeoides* L.																+		+		+	
*Carduus pycnocephalus* L.																	1	1		1	
*Holcus setiglumis* Boiss. et Reuter															+	+					
*Trifolium tomentosum* L.									+												
*Silene coeli-rosa* (L.) Godron															2						
*Oryzopsis miliacea* (L.) Asch. et Sch. subsp. *thomasii* (Duby) Nyman																				+	
*Onopordum nervosum* Boiss.											+										
*Mercurialis annua* L.																	+				
*Marrubium vulgare* L.																+					
*Lamarckia aurea* (L.) Moench					2																
*Geranium dissectum* L.				+																	
*Ecballium elaterium* (L.) A. Rich.																			1		
*Daucus carota* L. subsp. *maximus* (Desf.) Ball						**+**															
*Dactylis glomerata* L.										+											
*Calystegia sepium* (L.) R.Br.																1					

**Table 3 plants-13-00568-t003:** *Klaseo alcalae-Glebionetum discoloris ass. nova* (**holotypus** inv. 8).

N° inventory	1	2	3	4	5	6	7	**8**
N° cluster	KaG1	KaG2	KaG3	KaG4	KaG5	KaG6	KaG7	**KaG8**
Surface m^2^	2	2	2	2	2	2	2	**2**
Coverage (%)	90	85	85	95	95	95	95	**100**
Altitude (m)	170	135	66	20	20	60	65	**81**
Orientation	.	S	.	S	.	E	E	**E**
Inclination (%)	.	2	.	2	.	2	3	**2**
Average hight veg (cm)	70	60	60	40	50	50	60	**70**
**Characteristics association and upper units**								
*Glebionis discolor* (d’Urv.) Cano et al.	4	4	4	5	5	5	5	**5**
*Sinapis alba* L. subsp. *mairei* (H. Lindb.) Maire	2	2	+	2	1	2	.	**+**
*Andryala laxiflora* DC.	1	.	1	.	.	.	.	**+**
*Avena barbata* Pott ex Link	+	.	.	.	.	.	.	**.**
*Klasea alcalae* (Coss.) Cantó & Rivas Mart.	2	3	4	2	2	.	2	**4**
*Euphorbia boetica* Boiss.	2	2	.	.	.	.	.	**.**
*Echium plantagineum* L.	+	1	+		+	1	+	**1**
*Plantago lagopus* L.	1	.	.	1	.	.	.	**.**
*Avena sterilis* L.	+	+	+	+	.	+	+	**.**
*Calendula arvensis* L.	1	.	.	.	.	.	.	**.**
*Sisymbrium officinale* (L.) Scop.	+	1	.	.	.	.	.	**.**
*Lathyrus clymenum* L.	+	+	.	.	+	.	.	**.**
*Biscutella baetica* Boiss. & Reut.	+	.	.	.	.	.	.	**.**
*Crepis vesicaria* L.	.	1	+	1	.	1	+	**+**
*Urospermun picroides* (L.) Scop. ex Schmidt	.	1	1	+	.	.	+	**1**
*Hordeum murinum* L. subsp. *leporinum* Link	.	1	.	1	1	1	1	**.**
*Lavatera cretica* L.	.	+	.	.	.	.	.	**.**
*Reseda alba* L.	.	+	+	+	.	+	.	**1**
*Lathyrus latifolius* L.	.	+	.	.	.	.	.	**.**
*Galactites tomentosa* Moench	.	+	.	+	+	1	+	**.**
*Euphorbia helioscopia* L.	.	.	2	1	+	+	.	**.**
*Eruca vesicaria* (L.) Cav.	.	.	+	.	.	.	.	**.**
*Anagallis arvensis* L.	.	.	+	.	.	.	.	**.**
*Ononis viscosa* L.	.	.	+	.	.	.	.	**+**
*Malva neglecta* Wallr.	.	.	.	+	.	1	1	**2**
*Beta vulgaris* L. subsp. *maritima* (L.) Arcangeli	.	.	.	+	.	1	+	**1**
*Centaurea pullata* L.	.	.	.	+	1	.	.	**.**
*Glebionis coronaria* (L.) Cass. ex Spach	.	.	.	+	2	.	.	**+**
*Bromus diandrus* Roth	.	.	.	.	1	.	.	**.**
*Sonchus oleraceus* L.	.	.	.	.	.	.	+	**.**
*Anacyclus clavatus* (Desf.) Pers.	.	.	.	.	.	.	+	**.**
*Stipa capensis* Thumb.	.	.	.	.	.	.	.	**+**
*Mercurialis annua* L.	.	.	.	.	.	.	.	**+**
**Companion species**								
*Plantago bellardii* All.	+	.	.	.	.	.	.	**.**
*Genista equisetiformis* (Spach) Rivas Goday & Rivas Mart.	+	+	+	.	.	.	.	**.**
*Convolvulus althaeoides* L.	+	+	+	+	+	+	1	**.**
*Bituminaria bituminosa* (L.) C.H.Stirt.	+	1	1	1	.	.	1	**.**
*Lavandula multifida* L.	+	+	.	.	.	.	+	**.**
*Hyparrhenia sinaica* (Delile) Llauradó ex G. López	+	+	.	+	.	.	.	**.**
*Lotus edulis* L.	.	1	+	.	.	.	.	**.**
*Inula viscosa* (L.) Aiton	.	.	+	.	.	.	.	**.**
*Foeniculum vulgare* Miller	.	.	+	.	.	.	.	**.**
*Melica ciliata* L.	.	.	+	.	.	.	.	**.**
*Asteriscus maritimus* (L.) Less	.	.	.	1	.	.	.	**.**
*Asteriscus aquaticus* (L.) Less	.	.	.	.	.	+	2	**.**
*Asparagus albus* L.	.	.	.	+	+	.	.	**.**
*Calicotome villosa* (Poir.) Link	.	.	.	+	.	.	.	**.**
*Ricinus communis* L.	.	.	.	+	.	.	.	**.**
*Pallenis spinosa* (L.) Cass.	.	.	.	.	.	+	+	**+**
*Plantago afra* L.	.	.	.	.	.	.	1	**.**
*Lamarckia aurea* (L.) Moench	.	.	.	.	.	.	.	**+**

**Table 4 plants-13-00568-t004:** *Reichardio tingitanae-Glebionetum coronariae ass. nova* (**Holotypus** inv. 4).

N° inventory	1	2	3	**4**	5	6	7	8	9
N° Cluster	Rt-G1	Rt-G2	Rt-G3	**Rt-G4**	Rt-G5	Rt-G6	Rt-G7	Rt-G8	Rt-G9
Surface (m^2)^	4	4	4	**2**	2	4	4	4	4
Coverage (%)	85	100	85	**90**	100	100	70	90	95
Altitude (m)	4	8	50	**30**	80	2	2	3	20
Average hight veg. (cm)	30	25	40	**30**	40	35	30	40	45
**Characteristics association and upper units**									
*Glebionis coronaria* (L.) Cass. ex Spach	4	5	4	**4**	5	5	3	4	5
*Anacyclus clavatus* (Desf.) Pers.	2	3	+	**2**	2	2	2	1	+
*Hordeum murinum* L. *leporinum* Link	2	3	+	**1**	+		1	2	
*Bromus diandrus* Roth	1	2		**1**	1		1	1	2
*Echium plantagineum* L.			1	**2**	2	1		1	2
*Reichardia tingitana* (L.) Roth		+	1	**1**		1	+		1
*Malva neglecta* Wallr.	+		+	**1**		+		1	
*Chenopodium opulifolium* Schrader	1		+		1	1		1	
*Brassica repanda* (Willd.) DC. subsp. *confusa* (Emb. & Maire) Heywood			1				+	1	2
*Stipa capensis* Thumb.					2				1
*Lobularia maritima* (L.) Desv.			+		1	1		+	
*Sisymbrium officinalis* (L.) Scop.	1			**2**	1	1			
*Plantago serraria* L.	+	+		**+**					1
*Tetragonia tetragonoides* (Pall.) Kuntze		1		**2**					
*Plantago lagopus* L.		1		**2**					2
*Malva parviflora* L.		+							1
*Glebionis discolor* (d’Urv.) Cano et al.		+				1			
*Avena longiglumis* Durieu			1		1				
*Erodium malacoides* (L.) L’Hér.			+	**+**				+	
*Picris echioides* L.						+	+		
**Companion species**									
*Ricinus communis* L.	1	1		**1**	1	+			
*Lamackia aurea* (L.) Moench		+	+		1	+			
*Cynodon dactylon* (L.) Pers.		1		**1**					
*Euphorbia nicaensis* All.			1						1
*Piptatherum miliaceum* (L.) Cosson			+						+
*Foeniculum vulgare* Miller			+						+
*Silybum marianum* (L.) Gaertner	1	+		**+**					
*Polypogon viridis* (Gouan) Breistr.	+						+		
*Reseda lutea* L.					+		1		
*Amaranthus retroflexus* L.	1								
*Silene scabriflora* Brot. subsp. *tuberculata* (Ball) Talavera	+								
*Sonchus asper* (L.) Hill	+								
*Xanthium spinosum* L.	1								
*Anagallis arvensis* L.	+								
*Papaver rhoeas* L.	+								
*Rumex crispus* L.	1								
*Misopates orontium* (L.) Rafin.	+								
*Xanthium strumarium* L.	1								
*Lolium rigidum* Gaudin	1								
*Silene colorata* Poiret	+								
*Urtica membranacea* Poiret	+								
*Erodium laciniatum* (Cav.) Willd	2								
*Anagallis arvensis* L.		+							
*Xanthium strumarium* L.		1							
*Rumex crispus* L.		+							
*Lolium rigidum* Gaudin		1							
*Solanum nigrum* L.		1							
*Medicago polymorpha* L.		+							
*Plantago afra* L.			2						
*Crepis vesicaria* L.			+						
*Solanum nigrum* L.			+						
*Galactites tomentosa* Moench			+						
*Asphodelus fistulosus* L.			+						
*Inula viscosa* (L.) Aiton			+						
*Convolvulus althaeoides* L.				**1**					
*Tordilium maximum* L.				**+**					
*Anagallis arvensis* L.				**+**					
*Medicago coronata* (L.) Bartal.				**+**					
*Medicago arabica* (L.) Hudson				**+**					
*Leontodon salzmannii* (Schultz) Ball				**1**					
*Ononis ramosissima* Desf.				**1**					
*Plantago afra* L.				**1**					
*Convolvulus althaeoides* L.					+				
*Crepis vesicaria* L.					+				
*Misopates orontium* (L.) Rafin.					1				
*Euphorbia falcata* L.					1				
*Urospermum picroides* (L.) Scop.					+				
*Oxalis pes-capreae* L.					+				
*Mercurialis ambigua* (L.) Arcangeli						1			
*Anagallis arvensis* L.						+			
*Plantago afra* L.						1			
*Misopates orontium* (L.) Rafin.						+			
*Euphorbia falcata* L.						1			
*Brachypodium distachyon* (L.) Beauv.						+			
*Lolium rigidum* Gaudin						1			
*Silene nutans* L.						+			
*Medicago arabica* (L.) Hudson						+			
*Melilotus indica* (L.) All.						+			
*Avena sterilis* L.						+			
*Sonchus oleraceus* L.						+			
*Solanum nigrum* L.						+			
*Papaver somniferum* L.						+			
*Hirschfeldia incana* (L.) Lagr.-Foss.						+			
*Lotus collinus* (Boiss.) Heldr.						+			
*Anthyllis tetraphylla* L.						+			
*Xanthium strumarium* L.						+			
*Spergularia rubra* (L.) J. & C. Presl.						+			
*Andryala laxiflora* DC.							+		
*Lolium rigidum* Gaudin							1		
*Papaver rhoeas* L.							+		
*Crepis vesicaria* L.							+		
*Polycarpon tetraphyllum* L.							+		
*Campanula erinus* L.							+		
*Lactuca saligna* L.							+		
*Trifolium campestre* Schreber							+		
*Silene scabriflora* Brot.							+		
*Cynosurus echinatus* L.							+		
*Lagurus ovatus* L.							+		
*Xanthium strumarium* L.							+		
*Lactuca saligna* L.								1	
*Lolium rigidum* Gaudin								1	
*Avena sterilis* L.								+	
*Anagallis arvensis* L.								+	
*Crepis vesicaria* L.								+	
*Papaver somniferum* L.								+	
*Medicago murex* Willd.								+	
*Polypogon maritimus* Willd.									1
*Convolvulus arvensis* L.									1
*Lactuca saligna* L.									1
*Sonchus tenerrimus* L.									1

**Table 5 plants-13-00568-t005:** *Anacyclo valentianae-Glebionetum coronariae ass. nova* (inv. 1–11, **holotypus** inv. 2).

N° inventory	1	**2**	3	4	5	6	7	8	9	10	11
N° cluster	AvG1	**AvG2**	AvG3	AvG4	AvG5	AvG6	AvG7	AvG8	AvG9	AvG10	AvG11
Coverage (%)	95	**90**	80	100	80	85	100	100	100	50	100
Surface (m^2^)	40	**40**	40	25	25	40	8	4	10	10	20
Altitude (m)	149	**82**	285	68	82	90	256	231	175	228	120
Orientation											
Inclination (%)											
Average hight veg (cm)											
**Characteristics association and upper units**											
*Glebionis coronaria* (L.) Cass. ex Spach	5	**4**	4	5	4	4	3	4	5	2	5
*Sonchus oleraceus* L.	1	**1**	+	+	+	+	2	+			+
*Anacyclus valentinus* L.	*+*	**1**	+	+	+			+	+		+
*Echium creticum* L. subsp. *coincyanum* (Lacaita) R. Fernándes	1	**+**	+	1	1	+					2
*Hordeum murinum* L. subsp. *leporinum* (Link) Arcangeli	+	**+**	+	+	+		2	1	+		
*Eruca vesicaria* (L.) Cav.			1					2	1	3	+
*Moricandia arvensis* (L.) DC.		**+**			+	+			1		2
*Lavatera cretica* L.	1	**1**	1	1					2		1
*Rapistrum rugosum* (L.) All.	2		3	+	+	1					
*Calendula arvensis* L.					+	+	2	1	2		
*Avena sterilis* L.	1	**1**	+		1	1					
*Beta vulgaris* L.	1	**2**		3	1	1					
*Bromus rubens* L.	+	**1**				+				+	
*Carrichtera annua* (L.) DC.			+		+					2	1
*Lolium rigidum* Gaudin	1	**1**	+	1		1					
*Lophochloa pumila* (Desf.) Bor		**+**		+	+						
*Malva parviflora* L.							+	2			
*Malva sylvestris* L.		**+**		+		+					
*Medicago polymorpha* L.	1		+				2				
*Anacyclus clavatus* (Desf.) Pers.			1					2		+	
*Anagallis arvensis* L.			1			+					
*Papaver rhoeas* L.			+			+	1		+		
*Reichardia tingitana* (L.) Roth						+	+			+	
*Anchusa azurea* Mill.						1					
*Bartsia trixago* L.	+										
*Borago officinalis* L.			+								
*Brassica fruticulosa* Cirillo subsp. *cossoniana* (Boiss. & Reut.) Maire										3	
*Bromus diandrus* Roth										+	
*Bromus hordeaceus* L.	+										
*Bromus intermedius* Guss. subsp. *divaricatus* Bonnier & Layens						+					
*Bromus madritensis* L.	1	+	1								
*Bromus sterilis* L.	1										
*Calendula tripterocarpa* Rupr.			1								
*Centaurea melitensis* L.						1					
*Chenopodium album* L.		**1**		1	+						
*Chenopodium murale* L.					+				+		
*Glebionis discolor* (d’Urv.) Cano et al.			+		+						
*Diplotaxis erucoides* (L.) DC.								+			
*Emex spinosa* (L.) Campd.	1		+								
*Erodium chium* (L.) Willd.										1	
*Erodium malacoides* (L.) L’Hér.	1		3								
*Euphorbia serrata* L.						+					
*Galium tricornutum* Dandy		+									
*Halogeton sativus* (Loefl. ex L.) Moq.		+									
*Hedypnois cretica* (L.) Dum.Curs.	1										
*Lactuca serriola* L.				2	1						
*Lamium amplexicaule* L.							1				
*Moricandia moricandiodes* (Boiss.) Heywood					+						
*Papaver hybridum* L.					+		+				
*Plantago lagopus* L.	+					+					
*Reseda lutea* L. subsp. *lutea* L.			+								
*Rostraria pumila* (Desf.) Tzvelev					+						
*Scolymus hispanicus* L.	+										
*Scorpiurus muricatus* L.	+					+					
*Silene secundiflora* Otth in DC.			+								
*Silene vulgaris* (Moench) Garcke						1					
*Sisymbrium irio* L.								1			
*Sisymbrium officinale* (L.) Scop.		**2**		+							
*Sisymbrium orientale* L.							1				
*Sonchus tenerrimus* L.		**+**									
*Stipa capensis* Thunb.	+	**1**				+					
*Torilis arvensis* (Huds.) Link			+								
*Tripodion tetraphyllum* (L.) Fourr.	+										
*Urospermum picroides* (L.) Scop. ex F. W. Schmidt				+		+					
**Companion species**											
*Phalaris brachystachys* Link	1	**+**	+		+						
*Piptatherum miliaceum* (L.) Coss.	+	**+**		1		+					
*Convolvulus arvensis* L.		**+**	+	+		1					
*Convolvulus althaeoides* L.	+				1	+					
*Lamarckia aurea* (L.) Moench		**+**		+			1		+		
*Carduus bourgeanus* Boiss. & Reut.	1		+			2					
*Carthamus lanatus* L.					+	+					
*Cichorium intybus* L.	+										
*Cirsium vulgare* (Savi) Ten.						+					
*Misopates orontium* (L.) Rafin.							+				
*Onopordum macracanthum* Schousb.	+										
*Oxalis pres-caprae* L.					+						
*Verbascum pulverulentum* Vill.	+										
*Volutaria lippii* (L.) Maire									+		3
*Plantago afra* L.	+		+								
*Arisarum vulgare* Targ. -Tozz.	+										
*Artemisia barrelieri* Besser.	+										

**Table 6 plants-13-00568-t006:** *Centaureo napifoliae-Glebionetum coronariae ass. nova* (inv. 1–19 **holotypus** inv. 16). *Centaureo napifoliae-Glebionetum coronariae calenduletosum fulgidae subass. nova* (inv. 20–33 **holotypus** inv. 31).

N° inventory	1	2	3	4	5	6	7	8	9	10	11	12	13	14	15	**16**	17	18	19	20	21	22	23	24	25	26	27	28	29	30	**31**	32	33
N° cluster	11	12	13	14	15	16	18	20	22	25	29	33	34	35	36	**37**	38	41	43	39	40	17	21	23	24	26	30	31	32	42	**19**	27	28
Surface m^2^	15	15	15	10	15	20	25	20	15	20	10	20	20	10	15	**10**	10	15	15	12	10	15	20	25	12	20	12	12	20	15	**15**	15	12
Coverage (%)	100	100	85	85	100	90	90	90	100	95	85	100	100	95	90	**95**	95	100	90	100	95	95	95	80	95	100	90	100	90	100	**80**	90	90
Altitude (m)	4	10	35	85	17	70	177	50	27	44	356	10	140	83	85	**85**	404	339	63	492	471	42	100	115	32	7	323	193	151	315	**210**	217	534
Orientation	.	.	SW	.	.	.	.	.	.	.	.	.	.	.	.	**.**	.	S	.	.	.	.	.	.	.	.	SE	E	.	.	**SW**	.	.
Inclination (%)	.	.	3	.	.	.	.	.	.	.	.	.	.	.	.	**.**	.	15	.	.	.	.	.	.	.	.	20	10	.	.	**10**	.	.
Average hight veg (dm)	7	18	10	7	9	12	10	10	18	10	8	13	10	14	14	**13**	10	8	13	8	6	15	15	8	15	16	8	12	10	9	**12**	8	8
**Characteristics association and upper units**																																	
*Glebionis coronaria* (L.) Cass. ex Spach	5	5	4	4	5	5	4	4	4	4	1	+	5	5	+	**5**	4	5	4	4	5	5	4	4	4	5	1	1	1	5	**4**	1	4
*Glebionis discolor* (d’Urv.) Cano et al.	1	+		+	+		1	2	3	4	4	4	+		+	**+**				+		2	1	1	1	3	4	5	5	+	**+**	4	1
*Avena barbata* Pott ex Link		1	1	+	1	1	1		+	1	1		1			**1**					1	1	+	+	1	1							1
*Avena sterilis* L.	+	1	1	1	1	1	+	1		1				1		**2**	1	1	2	2	2	+	1	+	1	1		2	1	2	**2**	+	1
*Beta vulgaris* L.	2			1		1		+		1	2	1		+	1	**+**	1	+	+	1						1		1	1	2			
*Borago officinalis* L.	+		+					1	+		+	1		2	+	**+**	1		+								1	+	1		**+**	+	+
*Bromus madritensis* L.		+			+	+	1			1	1	+	+			**1**		2	1	1	1	+			1	1		1	1	2	**+**	1	
*Bromus sterilis* L.	+	1				+	1	+				1	+	+	+	**1**				1	1			+	1				+	1	**+**	1	1
*Galactites elegans* (All.) Nyman ex Soldano	+	1	2	+	1	1	1	1	2	1	1	1		1	1	**1**		2	2		2	1	2	2	1	1	1		2	1	**1**	1	1
*Lavatera cretica* L.	2	2	1	+	2	2	1		1	1	1	1	1		1	**+**	+	+				1		1	1	1	1	1	1	+		+	1
*Sonchus oleraceus* L.	+	1	+	+	+	1	+	1	+	+	1		1	+		**+**	1	+	+	1	1	+	+		+	+	1	1	1	1	**1**		
*Lotus ornithopodioides* L.	+		+				1	1		1				2		**1**	+					1	1	1	1	+	1	1	1			1	1
*Oxalis pes-caprae* L.	+	1	1	+	1	1		3	1				1	+	1	**1**			1	1		1	+	1		+		1	2		**+**		
*Erodium malacoides* (L.) L’Hér.	2	1			1	+	+	1			2			2	1	**1**		+	+	+		1	1	+								1	
*Melilotus sulcatus* Desf.			+	1	+	1	+	+		+			+										+		+	+	1					1	
*Mercurialis annua* L.	+	+	+	1	1	+		+	1	1		2	1	+					+			+	1		+		1	+	1				
*Anagallis foemina* Miller	1					+		+										+			1	+		+						+	**+**		
*Brassica fruticulosa* Cirillo	1		+	1	1	1				1			1											1		1		1	1		**+**		
*Sulla coronaria* (L.) Medik.	+	+	+			+	+	2		+								+					1			1	+		+		**+**		
*Anagallis arvensis* L.	+			1	+					+				+		**+**					+							+					
*Hordeum murinum* L. subsp. *leporinum* (Link) Arcangeli		+					+				1	+	+		2								+					+	+	+			
*Chenopodium murale* L.	2	+				1							+																				
*Solanum nigrum* L.	+	+			+																							+					
*Euphorbia helioscopia* L.			+	+		+		+		+					+	**+**													+	+	**1**		+
*Dasypyrum villosum* (L.) P. Candargy						1				1	+	+								1	+						+					1	1
*Medicago polymorpha* L.						1	+		+					+			1	1		+		+		1	1			1		1	**1**		
*Galium aparine* L.		+							+			+					1						+		+	+	+	+					
*Papaver rhoeas* L.				1									+	1		**+**				+	+							1		+	**+**	+	
*Urospermum picroides* (L.) Schmidt		+							1				1					+	+			+				+		+	1	+			
*Vicia sativa* L.		+				+	+	+						+		**+**		+				+		+				1		+		+	
*Sinapis arvensis* L.								1		+	+			+						+		+			2			+				1	
*Stipa capensis* Thunb.			1	+			+			+				+							+		+										
*Bromus hordeaceus* L.						+	+				+					**+**			+			+									**1**		
*Fumaria capreolata* L.					+				+				+												+								
*Trifolium nigrescens* Viv.										+	+														+	+		+				+	+
*Geranium molle* L.						+				+							1										1						
*Calendula arvensis* L.		+													+	**+**	1	+	+	2										+	**1**		
*Melilotus indicus* (L.) All.	1													2		**+**					+												
*Medicago truncatula* Gaertn.			+	2		+																					1						
*Sisymbrium officinale* (L.) Scop.	+					+							+																				
*Malva parviflora* L.				2	+	1							1		2					+		+											
*Stellaria media* (L.) Vill.									+																		+						+
*Hirschfeldia incana* (L.) Lagr.-Foss.				1			+	1															2			1							
*Capsella bursa-pastoris* (L.) Medik.				+	+																												
*Centaurea napifolia* L.										+		+	+			**+**	+							2									
*Diplotaxis erucoides* (L.) DC. subsp. *erucoides*															+			+	1	1	+									+			
*Plantago lagopus* L.						+	+								+							+	+										
*Echium plantagineum* L.														1	+					+			2	1	+					+			
*Silene gallica* L.				+						+													+		+							+	
*Reseda alba* L.										+																+		+				+	
*Lathyrus clymenum* L.			+																				+										
*Trifolium campestre* Schreber	+			+																													
*Medicago orbicularis* (L.) Bartal.	+																																
*Lophochloa cristata* (L.) Hill				1		1							+																				
*Sonchus asper* (L.) Hill								+		+			+																				
*Lolium rigidum* Gaudin							+						+	1																			
*Urtica membranacea* Poir. ex Savigny									+			+																					
*Misopates orontium* (L.) Rafin.				+																													
*Papaver dubium* L.				+																													
*Euphorbia terracina* L.					+																					+							
*Fumaria parviflora* Lam.					+																												
*Chenopodium album* L.						+																											
*Plantago lanceolata* L.					+																												
*Ridolfia segetum* (L.) Moris								1										+												+	**+**		
*Sinapis alba* L.						+										**+**	2			1	+												
*Gladiolus italicus* Mill.								+						+							+										**1**		
*Erodium cicutarium* (L.) L’Hér.							+																+										
*Vicia narbonensis* L.										+														+									
*Anacyclus tomentosus* (All.) DC.							+																								**1**		
*Centaurea solstitialis* L.								+																						1			
*Rumex conglomeratus* Murray									+																								
*Tragopogon porrifolius* L.										+																							
*Raphanus raphanistrum* L.												+																1					
*Lactuca serriola* L.														+																			
*Medicago scutellata* (L.) Mill.														+																			
*Reichardia picroides* (L.) Roth														+				+			+												
*Sherardia arvensis* L.														+			1	+		+							1			+			
*Scolymus grandiflorus* Desf.															+	**+**										+							
*Vulpia myuros* (L.) C.C.Gmel.																	+													+			
*Brassica nigra* (L.) W.D.J.Koch															1																		
*Silene vulgaris* (Moench) Garcke subsp. *tenoreana* (Colla) Soldano & F.Conti															+																		
*Hypochaeris radicata* L.																**+**																	
*Linum decumbens* Desf.																**+**																	
*Valerianella discoidea* (L.) Loisel.																	+																
*Coleostephus myconis* (L.) Cass. ex Rchb. f.																	+																
*Lathyrus aphaca* L.																		+												+			
*Hypochaeris achyrophorus* L.																		+												1			
*Senecio leucanthemifolius* Poir.																			+											+			
*Erodium botrys* (Cav.) Bertol.																		+		1													
*Anacyclus clavatus* (Desf.) Pers.																		+															
*Melilotus elegans* Salzm. ex Ser.																		1															
*Sisymbrium orientale* L.																		+															
*Calendula fulgida x arvensis*																				+	+	+	+	1	+	1	2	2	2		**2**	2	
*Calendula suffruticosa* Vahl subsp. *fulgida* (Raf.) Guadagno																														+	**1**	1	1
*Fedia graciliflora* Fisch. & C.A. Mey.																					+						+						
*Galium verrucosum* Huds. subsp. *verrucosum*																					+												
*Isatis tinctoria* L.																					+						+						
*Eruca vesicaria* (L.) Cav.																						+											
*Lotus edulis* L.																						+											
*Hedypnois rhagadioloides* (L.) F.W. Schmidt																							+										
*Silene fuscata* Brot.																														+			
*Chrysanthemum segetum* L.																															**+**		
*Bromus tectorum* L.																																+	
*Hyoscyamus albus* L.																																+	
**Companion species**																																	
*Carduus pycnocephalus* L.											1	1		+	2	**+**	1	+		2	1		+		1				1	+		1	
*Convolvulus althaeoides* L.	+	+	1	+	1	1				+				+		**+**				+	1	+	+										
*Cynodon dactylon* (L.) Pers.		+	1		+				+	+			1									+	+	+									
*Foeniculum vulgare* Miller subsp. *piperitum* (Ucria) Coutinho					+		+				+	+				**+**	+	+	+	+		+					1		+				+
*Ricinus communis* L.	+	+			+	+			+													+				+							
*Piptatherum miliaceum* (L.) Coss. subsp. *miliaceum*	+	+	+			+			+	+			+														+						
*Plantago afra* L.														+		**+**			+										+		**1**		
*Ecballium elaterium* (L.) A. Rich.										+			+						1		+												
*Emex spinosa* (L.) Campd.				+																													
*Eryngium campestre* L.																				+	+												
*Ferula communis* L.															+						+												
*Foeniculum vulgare* Miller									1																							+	
*Lavatera arborea* L.	1											+																					
*Matthiola tricuspidata* (L.) R. Br.	+																																
*Notobasis syriaca* (L.) Cass.														+				+												1			
*Phalaris brachystachys* Link				1																													
*Phalaris paradoxa* L.						+								+												+							
*Piptatherum miliaceum* (L.) Coss. subsp. *thomasii* (Duby) Freitag											1																						
*Silybum marianum* (L.) Gaertner																	+			+													
*Smyrnium olusatrum* L.																								+									
*Theligonum cynocrambe* L.																					+												
*Tordylium apulum* L.																	1	+											+	+			
*Acanthus mollis* L.																	+																
*Ammi visnaga* (L.) Lam.														+																			
*Asparagus albus* L.																**+**																	
*Asphodeliine lutea* (L.) Rchb.																					+												
*Brachypodium distachyum* (L.) Beauv.			+				+																+										
*Carthamus lanatus* L.	1																																
*Convolvulus arvensis* L.		+																															
*Cynara cardunculus* L.																						+											
*Daucus carota* L. subsp. *maximus* (Desf.) Ball																	+		+														

**Table 7 plants-13-00568-t007:** *Astragalo drupacei-Glebionetum coronariae ass. nova* (**holotypus** inv. 2).

N° Inventory	104	**105**	106	107	108	109	110	111	112
N° Cluster									
Coverage (%)	85	**95**	90	95	90	90	90	90	90
Surface (m^2^)	4	**4**	4	4	4	4	4	4	4
Altitude (m)	25	**20**	25	20	20	20	100	100	100
Orientarion	S	**SW**	S	E	.	.	.	.	.
Inclination (%)	6	**8**	20	20	.	.	.	.	.
Average hight veg (cm)	60	**75**	90	60	70	80	80	80	150
**Characteristics association and upper units**									
*Glebionis coronaria* (L.) Cass. ex Spach	4	**5**	1	1	5	5	5	5	5
*Hordeum murinum* L. subsp. *leporinum* (Link) Arcangeli	2	**1**	1	2	2	2	1	2	1
*Malva parviflora* L.	2	**2**	2	2	2	2	2	1	3
*Astragalus drupaceus* Orphan. ex Boiss.	1	**1**	1	1	+		1	1	1
*Sinapis alba* L. subsp. *dissecta* (Lag.) Bonnier	1	**2**	+			1	1	1	2
*Avena barbata* Pott ex Link	2	**1**			2	2	+	2	1
*Bromus diandrus* Roth *s.l.*	1	**1**	2	1	2	2		2	
*Lavatera arborea* L.			1	2	3	2	2	2	2
*Sonchus oleraceus* L.	1	**+**	+	1	1			1	+
*Reseda alba* L. subsp. *alba*	2	**1**				1	1	1	
*Urtica urens* L.			1	+		1	1		2
*Papaver rhoeas* L.	+	**+**						+	1
*Glebionis discolor* (d’Urv.) Cano et al.	2	**1**	4	5					
*Plantago lagopus* L.	2	**2**						2	
*Sisymbrium polyceratium* L.			2	2				2	
*Echium plantagineum* L.						1	1	1	
*Erodium moschatum* (L.) L’Hér.		**+**	1						
*Eruca vesicaria* (L.) Cav.	1	**+**							
*Euphorbia peplus* L.	+	**+**							
*Urtica pilulifera* L.			1						2
*Anagallis arvensis* L.					+		+		
*Hyoscyamus albus* L.			2	+					
*Lolium temulentum* L.			+						+
*Leontodon hispidus* L.	+	**1**							
*Sisymbrium irio* L.	+	**+**							
*Rapistrum rugosum* (L.) All.			1	1					
*Urospermum dalechampii* (L.) F.W. Schmidt	+	**+**							
*Rumex conglomeratus* Murray			1	1					
*Stipa capensis* Thunb.	1	**1**							
*Cynosurus echinatus* L.	+								
*Diplotaxis viminea* (L.) DC.					+				
*Diptychocarpus strictus* Trautv.			1						
*Echium arenarium* Guss.				+					
*Erodium maritimum* (L.) L’Hér.	+								
*Fumaria capreolata* L.			+						
*Galium tricornutum* Dandy									1
*Hedypnois cretica* (L.) Willd.				+					
*Lagurus ovatus* L.	+								
*Lepidium hirtum* (L.) Sm.	+								
*Anacyclus clavatus* (Desf.) Pers.									1
*Anthemis orientalis* (L.) Degen								1	
*Beta vulgaris* L. subsp *maritima* (L.) Arcang.								1	
*Bromus hordeaceus* L.						1			
*Calendula arvensis* L.								1	
*Cerinthe minor* L.					+				
*Papaver dubium* L.								+	
*Trifolium cherleri* L.								+	
*Urtica membranacea* Poir. ex Savigny	+								
*Vicia narbonensis* L.									+
*Sherardia arvensis* L.					+				
*Sonchus tenerrimus* L.						1			
*Stellaria media* (L.) Vill.							+		
**Companion species**									
*Oxalis pes-caprae* L.	2	**2**	2	2	1	2	+		
*Mercurialis annua* L.	1	**1**			+	+			1
*Rubia peregrina* L.			+		+		+	+	+
*Ricinus communis* L.			1	+	+	+			
*Silybum marianum* (L.) Gaertner						+	1		1
*Atriplex halimus* L.			1	+					
*Brachypodium distachyon* (L.) Beauv.	1	**1**							
*Plantago coronopus* L.			1	+					
*Asphodelus fistulosus* L.									2
*Atriplex glauca* L.				+					
*Carduus pycnocephalus* L.								2	
*Carduus tenuiflorus* Curtis	+								
*Convolvulus arvensis* L.					1				
*Nicotiana glauca* Graham				+					
*Tordylium apulum* L.		**+**							

**Table 8 plants-13-00568-t008:** Location and coordinates of the sampling points of the plant communities studied. The coordinates of the inventories of Portugual have not been included in this table because they have already been published previously in [10].

Association	N° Inventory	N° Cluster	Location	Latitude	Longitude	Association	N° Inventory	N° Cluster	Location	Latitude	Longitude
AvG	1	AvG1	Los Gallardos. A 1.7 Km	37.1568	−1.9503	CnG	1	11	Torrente Fiumarella (Reggio Calabria, Italy)	38.0223	15.6433
AvG	2	AvG2	Cuevas de Almanzora	37.285	−1.8902	CnG	2	12	Torrente Fiumarella (Reggio Calabria, Italy)	37.9819	15.6594
AvG	3	AvG3	Peñas Negras	37.0518	−2.0417	CnG	3	13	Capo dell’Armi (Reggio Calabria, Italy)	37.9537	15.6847
AvG	4	AvG4	Cuevas de Almanzora	37.2888	−1.8753	CnG	4	14	Saline Joniche (Reggio Calabria, Italy)	37.9544	15.7036
AvG	5	AvG5	Cuevas de Almanzora	37.2989	−1.8665	CnG	5	15	Sant’Elia (Reggio Calabria, Italy)	37.9317	15.7471
AvG	6	AvG6	Entre los lobos y La Muleria	37.3042	−1.7958	CnG	6	16	Sant’Elia (Reggio Calabria, Italy)	37.9307	15.7419
AvG	7	AvG7	Ferroliva	37.4133	−1.814	CnG	7	18	C.da Paliga, Melito P.S. (Reggio Calabria, Italy)	37.9323	15.7904
AvG	8	AvG8	Pozo de la Higuera	37.4372	−1.7424	CnG	8	20	Melito P.S. (Reggio Calabria, Italy)	37.9678	15.8036
AvG	9	AvG9	Restaurant Cuevas de Pulpí	37.367	−1.7337	CnG	9	22	Strada Lacco di Melito P.S. (R. Calabria, Italy)	37.9663	15.803
AvG	10	AvG10	Pozo de la Higuera	37.4374	−1.7405	CnG	10	25	San Carlo (Reggio Calabria, Italy)	37.9484	15.8834
AvG	11	AvG11	Barranco del tomate	37.2797	−1.9014	CnG	11	29	Fiumara dell’Amendolea (R. Calabria, Italy)	37.9736	15.8894
AdG	104	104	Acropolis (Atenas)	37.9727	23.7207	CnG	12	33	Catona (Reggio Calabria, Italy)	38.192	15.6431
AdG	105	105	Acropolis (Atenas)	37.9849	23.7067	CnG	13	34	Mota, Marinella di Palmi (R. Calabria, Italy)	38.3562	15.837
AdG	106	106	Thira (Santorini)	36.4016	25.473	CnG	14	35	Near Rosarno (Reggio Calabria, Italy)	38.4741	15.9514
AdG	107	107	Oia (Santorini)	36.4618	25.3753	CnG	15	36	Marina di Gioiosa (Reggio Calabria, Italy)	38.3123	16.3044
AdG	108	108	Kalamata	37.0279	22.1022	CnG	16	37	Exit from Siderno (Reggio Calabria, Italy)	38.2558	16.2831
AdG	109	109	Nafplio	37.5664	22.8196	CnG	17	38	Prox. Gerace (Reggio Calabria, Italy)	38.2677	16.2219
AdG	110	110	Epidauro	37.6995	23.1044	CnG	18	41	Strada Gerace-Cittanova Paso Zita (Reggio Calabria, Italy)	38.2904	16.1955
AdG	111	111	Micenas	37.7216	22.7436	CnG	19	43	Canolo (Reggio Calabria, Italy)	38.3154	16.201
AdG	112	113	Micenas	37.7161	22.7396	CnG	20	39	Canolo (Reggio Calabria, Italy)	38.3195	16.2042
CbG	1	CbG1	See coordinates	37.7105	−4.3025	CnG	21	40	Canolo-Agnana (Reggio Calabria, Italy)	38.2968	16.2524
CbG	2	CbG2	See coordinates	37.691	−4.3051	CnG	22	17	Siderno Superiore (Reggio Calabria, Italy)	38.2892	16.2736
CbG	3	CbG3	See coordinates	37.6903	−4.3048	CnG	23	21	Aeropuerto Reggio Calabria (Italy)	38.0754	15.6585
CbG	4	CbG4	See coordinates	37.6278	−4.1268	CnG	24	23	Prox. Siderno Superiore (Reggio Calabria, Italy)	38.2892	16.2736
CbG	5	CbG5	See coordinates	38.0361	−4.2196	CnG	25	24	Strada Catania-Giardini (Sicily, Italy)	37.3768	14.7287
CbG	6	CbG6	See coordinates	37.8271	−4.0468	CnG	26	26	A 14 Km de Gela (Sicily, Italy)	37.1671	14.2851
CbG	7	CbG7	See coordinates	37.9283	−4.061	CnG	27	30	A 14 Km de Gela (Sicily, Italy)	37.1671	14.2851
CbG	8	CbG8	See coordinates	37.9285	−4.0646	CnG	28	31	Monte Gibliscemi (Sicily, Italy)	37.2106	14.2712
CbG	9	CbG9	See coordinates	37.9381	−4.0662	CnG	29	32	Carretera Etna-Catania (Sicily, Italy)	37.5073	14.2284
CbG	10	CbG10	See coordinates	37.9423	−4.0787	CnG	30	42	Capodasa-Estrada 117 a Palermo (Sicily, Italy)	37.5111	14.2174
CbG	11	CbG11	See coordinates	37.9905	−4.1237	CnG	31	19	Carretera Etna-Catania (Sicily, Italy)	37.5476	14.3959
CbG	12	CbG12	See coordinates	38.024	−4.128	CnG	32	27	Estrada Provincial-Etna-Catania (Sicily, Italy)	37.566	14.4042
CbG	13	CbG13	See coordinates	38.0384	−4.1142	CnG	33	28	Autostrada Catania (Sicily, Italy)	37.4691	14.8726
CbG	14	CbG14	See coordinates	38.0314	−4.0659	KaG	1	KaG1	Equestrian Club Proximities	36.7017	−4.3875
CbG	15	CbG15	See coordinates	38.0356	−4.0375	KaG	2	KaG2	Proximities Mediterranean Highway/Almendrales Road	36.7417	−4.4056
CbG	16	CbG16	See coordinates	38.0242	−3.9499	KaG	3	KaG3	Proximities of Casa Pedro and Ana Restaurant	36.7375	−4.4042
CbG	17	CbG17	See coordinates	37.8719	−4.1692	KaG	4	KaG4	Sierra del Coo-Abandoned Garden	36.7264	−4.4042
CbG	18	CbG18	See coordinates	37.8736	−4.1688	KaG	5	KaG5	Gibralfaro Road. Proximities Mount Victoria	35.7264	−4.4042
CbG	19	CbG19	See coordinates	37.8802	−4.1895	KaG	6	KaG6	Climb to Gibralfaro	36.725	−4.4056
CbG	20	CbG20	See coordinates	37.9168	−4.2078	KaG	7	KaG7	Climb to Gibralfaro	36.725	−4.4056
CbG	21	CbG21	See coordinates	37.9266	−4.2072	KaG	8	KaG8	Proximities Arroyo Toquero	36.7361	−4.4056
RtG	1	Rt-G1	Rambla prox. Guadalfeo (Lobres, Granada)	36.7698	−3.5624						
RtG	2	Rt-G2	Rambla del Guadalfeo	36.7703	−3.55						
RtG	3	Rt-G3	Prox. Lobres	36.7744	−3.5706						
RtG	4	Rt-G4	Lobres (Estación Eléctrica)	36.6476	−3.552						
RtG	5	Rt-G5	prox. Casa Rosa (Salobreña, Granada)	36.7396	−3.5893						
RtG	6	Rt-G6	Prox. Playa Granada (Motril, Granada)	36.7249	−3.5256						
RtG	7	Rt-G7	Desembocadura del Guadalfeo (Motril)	36.5278	−3.5771						
RtG	8	Rt-G8	Urbanizaciones de Salobreña (Granada)	36.7355	−3.5857						
RtG	9	Rt-G9	Prox. Torrenueva (Granada)	36.7129	−3.4944						

**Table 9 plants-13-00568-t009:** Material studied for Italy and Greece.

Italy-Greece	Rel. Cluster
*Malvo parviflorae-Chrysanthemetum coronarii* (Brullo et al., 2001) [25] Reggio Calabria (Italy)	1–10
Own relevés Sicilia and Reggio Calabria (Italy)	11–43
*Malvo parviflorae-Chrysanthemetum coronarii* (Ferro, 1980, table 13) [27]	44–62
*Lavatero creticae-Chrysanthemetum coronarii* (Ferro, 2004, table 5) [28]	63–72
*Hordeo-Centauretum macracanthae* (Brullo, 1983, table 10) [26]	73–77
*Chrysanthemo-Silybetum mariani* (Brullo, 1983, table 11) [26]	78–82
*Malvo parviflorae-Chrysanthemetum coronarii* (Brullo, 1983, table 1) [26]	83–92
Own relevés (Cano) (Greece)	93–101

**Table 10 plants-13-00568-t010:** Variables used in the statistical analysis.

Variable	Min	Max	Mean	Stand. Dev.	KMO
N	0.98	3.30	1.45	0.34	0.94
Ios4	0.02	0.16	0.08	0.02	0.80
Ios3	0.09	1.21	0.52	0.22	0.79
PREC_TOTAL	257.35	870.65	553.23	173.97	0.78
Tp	1210.33	2048	1529.08	230.45	0.78
pH_CaCl	5.58	7.63	7.00	0.46	0.78
Io	1.82	6.20	3.60	0.94	0.76
Ios2	0.06	1.09	0.37	0.28	0.76
pH_H_2_O	6.16	8.30	7.63	0.48	0.73
Coar_Frag.	8.30	26.84	14.25	3.81	0.73
PE	591.14	825.48	666.15	61.19	0.72
CaCO_3_	31.28	507.86	217.01	94.77	0.72
Bulk_Dens	0.85	1.47	1.21	0.14	0.70
IH	−59.53	37.94	−17.63	22.42	0.68
PEs	94.87	132.23	104.64	9.78	0.68
Iar	0.72	2.47	1.32	0.43	0.68
Ioe	0.40	1.38	0.82	0.22	0.68
AWC	0.07	0.14	0.11	0.01	0.67
CN	8.77	15.59	11.61	1.44	0.65
K	153.31	612.03	337.32	100.65	0.60
Sand	10.85	64.45	30.10	8.83	0.58
Clay	11.98	50.44	29.71	5.93	0.58
Itc	296.67	451.67	376.48	39.31	0.55
Ic	12.83	16.57	14.58	1.02	0.54

## Data Availability

Data are contained within the article.

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
