# Peer review of "Ecological and Syntaxonomic Analysis of the Communities of Glebionis coronaria and G. discolor (Malvion neglectae) in the European Mediterranean Area"

_plants, 2024, doi:10.3390/plants13050568_

Round 1
Reviewer 1 Report (Previous Reviewer 2)
Comments and Suggestions for Authors
The paper has greatly improved with the accurate revision of the authors: I would only like to signal a small erro in references 52 and 53 where Falleni should read Fanelli
Author Response
REVISOR 1
The paper has greatly improved with the accurate revision of the authors: I would only like to signal a small erro in references 52 and 53 where Falleni should read Fanelli
REPLY TO REVIEWER 1
Dear reviewer
Thanks for your comments. We have made stylistic corrections to the English and Falleni is corrected by Fanelli, also because references 52 and 52 were repeated, they have also been corrected, and consequently it has been renumbered in the text, the reference now being 42

Reviewer 2 Report (Previous Reviewer 3)
Comments and Suggestions for Authors
May be published with minor stylistic changes
Comments on the Quality of English LanguageMay be published with minor stylistic changes
Author Response
REVISOR 2
May be published with minor stylistic changes. Comments on the Quality of English Language
REPLY TO REVIEWER 2
Dear reviewer
Thank you for your comments on the small stylistic corrections of the English language. We have made style corrections that you have requested.

Reviewer 3 Report (Previous Reviewer 4)
Comments and Suggestions for Authors
Comments and suggestions are given at the attached file.

Author Response
REVISOR 3
Comments and suggestions are given at the attached file.
REPLY TO REVIEWER 3
Dear reviewer
Thank you for the comments expressed in the text. We have taken into consideration all the comments you have given us, and the requested corrections have been made, you can see the corrections in red. In the specific case of eliminating "et al.", it was done when references were blocked, but if "et al." corresponded to the authors of a syntaxon, it is not possible to delete it, because this would modify the authorship of the syntaxons.

This manuscript is a resubmission of an earlier submission. The following is a list of the peer review reports and author responses from that submission.
Round 1
Reviewer 1 Report
Comments and Suggestions for Authors
Review - Phytosociological study of Mediterranean grasslands
The author present results of a phytosociological study of Mediterranean grasslands. The manuscript contains a lot of information collected in relevés and the authors used appropriate methods to analyze and interpret the data but further analysis (see comments below) might be considered.
The authors interpreted the results as providing evidence that there needs to be a ‘new’ taxonomy for grasslands in the countries where the study was collected.
I provide a few comments below on how the manuscript might be improved but there are basic problems that must be addressed. What were the reasons for doing the research? Why is it important to conduct research of this type in a historical as well as a modern context? What will be learned from the research that will provide a better understanding of the ecological functions of the different plant communities in the different countries and how will that benefit land management?
The language could be improved by using less jargon and attempting to resent the results in a way that makes it more understandable to people who are not familiar with the phytosociological terminology.
Authors might consider using the past-tense throughout as they are reporting on research that has been completed. An example is the 2nd line of the Abstract.
Lines 26. The language needs to be changes to something like: ‘Identification of grassland communities is important because differences in species assemblages are indicative of differences in substrate nutrient conditions.’ The current text is not clear. For example, what is the meaning of ‘Grasslands whose knowledge is fundamental’? Grasslands don’t have knowledge and ‘fundamental’ to what?
The Abstract provide results but the context for the material is not developed. Why should someone care that the edaphic characteristics are close to the two communities noted? Also, previously we are told that the research resulted in six new associations and one subassociation. How does that match with the later comment that something is related to two communities?
Lines 35-38. More information is need on why studies of this type are important, and in the following sentences, more background is needed on why nitrophilous grasslands are special.
Line 42. Delete ‘already’
A lot of the Introduction includes material that is not provided in any context. Lines 151-154 provide some context on why the study was done and why it is important. More information is needed, however, on why the results of the study will be important to land managers.
Why were the studies conducted (i.e., what did they hope to learn from the research that was not already known and what is the relevance). It is important to present this research in a context that is relevant to one or more ecological or sociological issues. Most readers of the journal will not be familiar with the phytosociological terminology – which was and is mostly used in a European context.
Results are presented in great details but without a context for the research (see earlier comments) it is impossible for someone who is not familiar with the vegetation of the region to understand them. Also, give the extent of most of the tables, the authors should consider presenting them as Supplemental material – thus enabling them to focus on the main points of the research. The many pages of the phytosociological table, for example, needs to be presented as supplemental material and only the important parts of the Table presented in the manuscript.
The author suggest 4 new associations and the material that follows in lines 238-294 attempts to present the relevance of the next taxonomy to what has previously been done. The goal of that effort is appropriate but much of what is presented in those lines seems to be more background information, which could have been presented in some way in the Introduction to justify the reasons for doing the research. More emphasis in the Discussion should be given to how the new taxonomy will benefit land managers.
The Conclusion section gives some insight, perhaps, into why the study was conducted.
I do not follow the European phytosociological literature and thus am not capable of knowing if all of the relevant material is included.
The authors should also consider doing a further analysis that shows the relationships (i.e., an ordination) between the soil characteristics and the associations that were identified. Presenting the result of the soil research in a Table does not seem to be adequate.
Comments on the Quality of English LanguageSee comments above to the authors about the language.
Reviewer 2 Report
Comments and Suggestions for Authors
The papers presents a study of the communities with Glebionis coronaria and Glebionis discolor in the whole mediterraenan basin. The paeper is triking for the amnount of material presented and the analysis is sound, but the presentation is quite a mess.
The main problem is that hthe authors make reference to the literature without shotrly presenting the content of this literature. In particular all the paper is founded on the morphometric study of Glebionis which is not presented at all and makes the paper difficult to udnerstand. Consider also that not all the readeer may agree with your taxonomy, and therefore you need to support your taxonomical decsions (although of course they are substantiated in the paper on the morphometry op Glebionis but again you have not the last word on the subject9
Moreover the discussion of nomebnclature is often incorrect or not completely correct.
Finally the english is correct but often the style is very poor
It seems in general that the paper was prepared quite in a hurry, and i suggest to speend a little more time in polishing the presentation since thi paper is very important.
Finally there are often ovious considerations that are better deleted.
Concerning the figure a map fo the distribution of the many associations dscribed, wmany of which are new is not only useful but I dare say mandatory in this study.
in t he attached file you can read minor observations

Comments on the Quality of English LanguageReviewer 3 Report
Comments and Suggestions for Authors
I would recommend doing a combined analysis with neighboring countries or with neighboring phytochorions, with neighboring phytogeographic regions. This would be useful for understanding the dynamics and distribution of the studied communities. In addition, I advise you to do a more detailed statistical analysis using climatic and other data, for example from a gene bank. In this case, the work would be more interesting for a wide range of readers!
Reviewer 4 Report
Comments and Suggestions for Authors
Please see attachment.

Comments on the Quality of English Languagenone